# Can we globally optimize cross-validation loss? Quasiconvexity in ridge regression

**William T. Stephenson**[*]
MIT
wtstephe@mit.edu

**Zachary Frangella**
Cornell
zjf4@cornell.edu

**Madeleine Udell**
Cornell
udell@cornell.edu

**Tamara Broderick**
MIT
tbroderick@mit.edu

## Abstract

Models like LASSO and ridge regression are extensively used in practice due to their interpretability, ease of use, and strong theoretical guarantees. Cross-validation (CV) is widely used for hyperparameter tuning in these models, but do practical optimization methods minimize the true out-of-sample loss? A recent line of research promises to show that the optimum of the CV loss matches the optimum of the out-of-sample loss (possibly after simple corrections). It remains to show how tractable it is to minimize the CV loss. In the present paper, we show that, in the case of ridge regression, the CV loss may fail to be quasiconvex and thus may have multiple local optima. We can guarantee that the CV loss is quasiconvex in at least one case: when the spectrum of the covariate matrix is nearly flat and the noise in the observed responses is not too high. More generally, we show that quasiconvexity status is independent of many properties of the observed data (response norm, covariate-matrix right singular vectors, and singular-value scaling) and has a complex dependence on the few that remain. We empirically confirm our theory using simulated experiments.

## 1 Introduction

Linear models, including LASSO and ridge regression, are widely used for data analysis across diverse applied disciplines. Linear models are often preferred since they are straightforward to apply in various senses. In particular, (1) their parameters are readily interpretable. (2) They have strong theoretical guarantees on quality. And (3) standard optimization tools are often assumed to find useful parameter and hyperparameter values. Despite their seeming simplicity, though, mysteries remain about the quality of inference in linear models. Consider cross-validation (CV) [Stone, 1974, Allen, 1974], the de facto standard for hyperparameter selection across machine learning methods [Musgrave et al., 2020]. CV is an easy-to-evaluate proxy for the true out-of-sample loss. Is it a good proxy? [Homrighausen and McDonald, 2014, 2013, Chetverikov et al., 2020, Hastie et al., 2020, Patil et al., 2021] give conditions under which the global minimum of the CV loss (possibly with some mild corrections) matches the optimum of the out-of-sample loss in LASSO and ridge regression. To complete the picture, we must understand whether standard methods for minimizing the CV loss find a global minimum.

It would be easy to find a unique minimum of the CV loss if the CV loss were convex. Alas (though perhaps unsurprisingly), we show below that in essentially every case of interest the CV loss is not convex. Indeed, the usual introductory cartoon of CV loss (left panel of Fig. 1; see also Fig. 5.9 of Hastie et al. [2017] or Fig. 1 of Rad and Maleki [2020]) is not convex. But the cartoon CV loss still exhibits a single global minimum and would be easy to globally minimize with popular approaches like gradient-based methods [Do et al., 2007, Maclaurin et al., 2015, Pedregosa, 2016, Lorraine et al., 2020] or grid search [Bergstra and Bengio, 2012, Pedregosa et al., 2011, Hsu et al., 2003]. Indeed, a

---

[*]Alternate email: wtstephe@gmail.com

35th Conference on Neural Information Processing Systems (NeurIPS 2021).

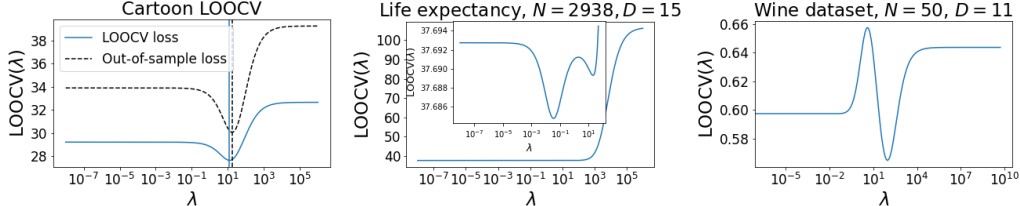

Figure 1: (*Left*): Idealized illustration of the leave-one-out CV loss $\mathcal{L}$ (blue) and the true out-of-sample loss (black). The minimizer of each curve is marked with a vertical line of the corresponding color. (*Center*): CV loss for a life-expectancy prediction problem after some standard data pre-processing (Condition 1 of Section 2). (*Right*): CV loss for wine-quality prediction problem on a subset of $N = 50$ data points after standard data pre-processing (Condition 1 of Section 2).

more plausible possibility (which holds for the typical cartoon CV loss) is that the CV loss might be *quasiconvex*; i.e. its level sets are convex. The benefit of quasiconvexity is that, in one dimension, any local optimum is a global optimum.

Unfortunately, this cartoon need not hold in general, even in simple models like $\ell_2$-regularized linear regression (i.e. ridge regression). Consider minimizing leave-one-out CV (LOOCV) loss as a function of the ridge regularization parameter; we denote this loss by $\mathcal{L}$. Wilson et al. [2020, Fig. 1] detail a simulated example in which $\mathcal{L}$ can be non-quasiconvex. We first demonstrate that $\mathcal{L}$ can be non-quasiconvex in real-data examples; see the middle and right panel of Fig. 1, which we describe in detail in Section 3. We next characterize which aspects of the covariate matrix and observed responses affect quasiconvexity. We prove that the norm of the responses, the scale of the singular values of the covariate matrix, and the right singular vectors of the covariate matrix all have no effect on the quasiconvexity of $\mathcal{L}$. While this result places substantial constraints on what drives the quasiconvexity of $\mathcal{L}$, we show that the quasiconvexity of $\mathcal{L}$ is unfortunately still a complex function of the remaining quantities. Our third contribution is to prove conditions under which $\mathcal{L}$ is guaranteed to be quasiconvex. In particular, we show that if (1) the covariate matrix has a singular value spectrum sufficiently close to uniform, (2) the least-squares estimator fits the training data sufficiently well, and (3) the left singular vectors of the covariate matrix are sufficiently regular, then $\mathcal{L}$ is guaranteed to be quasiconvex. While the conditions of our theory are deterministic, we show that they have natural probabilistic interpretations; as a corollary to our theory, we demonstrate that many of our conditions are satisfied either empirically or theoretically by well-specified linear regression problems with i.i.d. sub-Gaussian covariates and moderate signal-to-noise ratios. Through empirical studies, we validate the conclusions of our theory and the necessity of our assumptions.

## 2 Setup and notation

For $n \in \{1, \ldots, N\}$, we observe covariates $x_n \in \mathbb{R}^D$ and responses $y_n \in \mathbb{R}$. We are interested in learning a linear model between the covariates and responses, $\langle x_n, \theta \rangle \approx y_n$, for some parameter $\theta \in \mathbb{R}^D$. In ridge regression, i.e. $\ell_2$-regularized linear regression, we take some $\lambda > 0$ and estimate:

$$\hat{\theta}(\lambda) := \arg\min_{\theta \in \mathbb{R}^D} \sum_{n=1}^{N} (\langle x_n, \theta \rangle - y_n)^2 + \frac{\lambda}{2} \|\theta\|_2^2. \tag{1}$$

The regularization parameter $\lambda$ is typically chosen by minimizing the cross-validation (CV) loss. Here we study the leave-one-out CV (LOOCV) loss:

$$\mathcal{L}(\lambda) := \sum_{n=1}^{N} \left( \langle x_n, \hat{\theta}^{\backslash n}(\lambda) \rangle - y_n \right)^2, \tag{2}$$

where $\hat{\theta}^{\backslash n}(\lambda)$ is the solution to Eq. (1) with the $n$th datapoint left out.

Let the covariate matrix $X \in \mathbb{R}^{N \times D}$ be the matrix with rows $x_n$, and let the vector $Y \in \mathbb{R}^N$ be the vector with entries $y_n$. We consider the low to modest-dimensional case where $D < N$ and assume the covariate matrix $X$ is full-rank. We further assume $X$ and $Y$ have undergone standard data pre-processing, as described next.

**Condition 1.** *$Y$ is zero-mean, and $X$ has zero-mean, unit variance columns. Equivalently, where $\mathbf{1} \in \mathbb{R}^N$ is the vector of all ones, $\mathbf{1}^T Y = 0$ and $X^T \mathbf{1} = \mathbf{0} \in \mathbb{R}^D$ and for all $d = 1, \ldots, D$, $\sum_{n=1}^N x_{nd}^2 = N$.*

Preprocessing $X$ and $Y$ to satisfy Condition 1 represents standard best practice for ridge regression. First, using an unregularized bias parameter in Eq. (1) and setting $Y$ to be zero-mean are equivalent; we choose to make $Y$ zero-mean, as it simplifies our analysis below. The conditions on the covariate matrix $X$ are important to ensure the use of $\ell_2$-regularization is sensible. In particular, Eq. (1) penalizes all coordinates of $\theta$ equally. If e.g. some columns of $X$ are measured in different scales or are centered differently, this uniform penalty will be inappropriate.

# 3 LOOCV loss is typically not convex and need not be quasiconvex

If the LOOCV loss $\mathcal{L}$ were convex or quasiconvex in $\lambda$, then any local minimum of $\mathcal{L}$ would be a global minimum, and we could trust gradient-based optimization methods or grid search methods to return a value near a global minimum. We next see that unfortunately $\mathcal{L}$ is typically not convex and is often not even quasiconvex. First we show that, in essentially all cases of interest, $\mathcal{L}$ is *not* convex.

**Proposition 1.** *If $\lambda = \infty$ is not a minimum of $\mathcal{L}$, then $\mathcal{L}$ is not a convex function.*

*Proof.* For the sake of contradiction, assume $\mathcal{L}$ is convex and $\lambda = \infty$ is not a minimum of $\mathcal{L}$. This implies that there is some maximal $\lambda^* < \infty$ such that $\mathcal{L}'(\lambda^*) = 0$. Let $\delta := \mathcal{L}'(\lambda^* + 1)$. By convexity, $\mathcal{L}'' \geq 0$, so we know that $\delta > 0$ and that for $\lambda \geq \lambda^* + 1$, we have $\mathcal{L}'(\lambda) \geq \delta$. Thus for $\lambda \geq \lambda^* + 1$, we have $\mathcal{L}(\lambda) \geq \delta(\lambda - \lambda^* - 1)$. So $\lim_{\lambda \to \infty} \mathcal{L}(\lambda) = \infty$. However, inspection of $\mathcal{L}$ shows $\lim_{\lambda \to \infty} \mathcal{L}(\lambda) = \sum_{n=1}^N y_n^2 < \infty$, which is a contradiction. $\square$

We say that the result covers essentially all cases of interest: if $\mathcal{L}$ continues to decrease as $\lambda \to \infty$, then there is so little signal in the data that the zero model $\theta = \mathbf{0} \in \mathbb{R}^D$ is the optimal predictor according to LOOCV.

Although $\mathcal{L}$ is generally not convex, $\mathcal{L}(\lambda)$ might still be easy to optimize if it satisfies an appropriate generalized notion of convexity. To that end, we recall the definition of quasiconvexity.

**Definition 1.** *A function $f : \mathbb{R}^p \to \mathbb{R}$ is* quasiconvex *if its level sets are convex.*

In one dimension (i.e. $p = 1$ in Definition 1), quasiconvexity guarantees that any local optimum is a global optimum, just as convexity does. This property is often the key consideration in practical optimization algorithms. Moreover, it is not unreasonable to hope that the CV loss is quasiconvex: typical illustrations of the CV loss are not convex but are quasiconvex; see e.g. Hastie et al. [2015, Fig. 5.9], Rad and Maleki [2020, Fig. 1], or the left panel of Fig. 1. Illustrations of the out-of-sample loss are also typically quasiconvex; see e.g. Fig. 3.6 of Bishop [2006].

Unfortunately, we next demonstrate that the CV loss derived from real data analysis problems can be non-quasiconvex. Our first dataset contains $N = 2{,}938$ observations of life expectancy, along with $D = 20$ covariates such as country of origin or alcohol use; see **??** for a full description. In this case, after pre-processing according to Condition 1, $\mathcal{L}$ for the full dataset is quasiconvex. But now consider some additional standard data pre-processing. Practitioners often perform principal component regression (PCR) with the aim of reducing noise in the estimated $\theta$. That is, they take the singular value decomposition of $X = USV$ and produce an $N \times R$ dimensional covariate matrix $X'$ by retaining just the top $R$ singular values of $X$: $X' = U_{\cdot, :R} S_{:R}$. With this pre-processing, the resulting LOOCV curve $\mathcal{L}$ is non-quasiconvex for many values of $R$; in the center panel of Fig. 1 we show one example for $R = 15$.

Our second dataset consists of recorded wine quality of $N = 1{,}599$ red wines. The goal is to predict wine quality from $D = 11$ observed covariates relating to the chemical properties of each wine; see **??** for a full description. We find that subsets of this dataset often exhibit non-quasiconvex $\mathcal{L}$. In the right panel of Fig. 1, we show $\mathcal{L}$ for a subset of size $N = 50$. We see that this plot contains at least two local optima, with substantially different values of $\lambda$ and substantially different values of the loss. A gradient-based algorithm initialized sufficiently far to the left would not find the global optimum, and grid search without sufficiently large values would not find the global optimum.

Now we know that $\mathcal{L}$ can be non-quasiconvex for real data. Given the difficulty of optimizing a function with several local minima, we next seek to understand *when* $\mathcal{L}$ is quasiconvex or not.

## 4   What does the quasiconvexity of $\mathcal{L}$ depend on?

We have seen that $\mathcal{L}$ can be quasiconvex or non-quasiconvex, depending on the data at hand. If we could determine the quasiconvexity of $\mathcal{L}$ from the data, we might better understand how to efficiently tune hyperparameters from the CV loss. In what follows, we start by showing that the quasiconvexity of $\mathcal{L}$ is, in fact, independent of many aspects of the data (Proposition 2). We will see, however, that the dependence of quasiconvexity on the remaining aspects (though they are few) is complex.

A linear regression problem has a number of moving parts. The response vector $Y$ may be an arbitrary vector in $\mathbb{R}^N$, and the covariate matrix $X$ can be written in terms of its singular values and left and right singular vectors. More precisely, let $X = U \operatorname{diag}(S) V^T$ be the singular value decomposition of the covariate matrix $X$, where $U \in \mathbb{R}^{N \times D}$ is an $N \times D$ matrix with orthonormal columns, $S \in \mathbb{R}^D$ is a vector with positive entries, and $V \in \mathbb{R}^{D \times D}$ is an orthonormal matrix. Note we use the "compact" singular value decomposition, where $U$ is a $N \times D$ matrix, rather than a full $N \times N$ matrix. With this notation in hand, we can identify aspects of the problem that do not contribute to quasiconvexity in the following result, which is proved in **??**.

**Proposition 2.** *The quasiconvexity of $\mathcal{L}$ is independent of*

1. *the matrix of right singular vectors, $V$,*

2. *the norm of the responses, $\|Y\|_2$, and*

3. *the scaling of the singular values (i.e. changing $S$ into $S/c$ for $c \in \mathbb{R}_{>0}$),*

*in the sense that altering any of these quantities does not change whether or not $\mathcal{L}$ is quasiconvex.*

**Remark 1.** *Assume Condition 1 holds. Then by Proposition 2, without loss of generality we may (and do) assume that $V = I_D$ and that $Y$ is a vector on the unit $(N-1)$-sphere.*

Recall $X$ has zero-mean columns by pre-processing the data (Condition 1). By Proposition 2, we assume without loss of generality $V = I_D$. Thus, the columns of $X$ have zero mean when $U^T \mathbf{1} = 0$, where $\mathbf{1} \in \mathbb{R}^N$ is the vector of all ones. Also, while Remark 1 notes that we can consider $Y$ to be on the unit $(N-1)$-sphere, note that the condition $\mathbf{1}^T Y = 0$ from Condition 1 allows us to further constrain $Y$ to be parameterized by a vector on the unit $(N-2)$-sphere. Hence the quasiconvexity of $\mathcal{L}$ depends on three quantities: (1) the matrix of left singular vectors, $U$, an orthonormal matrix with $\mathbf{1} \in \mathbb{R}^N$ in its left null-space, (2) the (normalized) vector $Y$ which sits on the unit $(N-2)$-sphere, and (3) the (normalized) singular values.

Now that we know the quasiconvexity of $\mathcal{L}$ depends on only three quantities, we might hope that quasiconvexity would be a simple function of the three. To investigate this dependence, we consider the case of $N = 3$ and $D = 2$, since this case is particularly easy to visualize. To see why it is easy to visualize, first note that $Y$ is a three-dimensional vector; thus by our discussion above, we can parameterize $Y$ by a vector on the unit circle (i.e. a scalar between $0$ and $2\pi$). Second, note that the matrix of left singular vectors $U$ is parameterized by two orthonormal vectors, $U_{\cdot 1}$ and $U_{\cdot 2}$, each on the unit 2-sphere. As both vectors must be orthogonal to $\mathbf{1} \in \mathbb{R}^3$, we can parameterize $U_{\cdot 1}$ and $U_{\cdot 2}$ by two orthonormal vectors on the unit circle. We parametrize $U_{\cdot 1}$ by a scalar that determines $U_{\cdot 2}$ up to a rotation of $U_{\cdot 2}$ by $\pi$. We fix a rotation for $U_{\cdot 2}$ relative to $U_{\cdot 1}$ so that, for fixed singular values $S$, the quasiconvexity of $\mathcal{L}$ is parameterized by two scalars.

Fig. 2 is the resulting visualization. Precisely, to make Fig. 2, we fix an orientation between $U_{\cdot 1}$ and $U_{\cdot 2}$: here, a rotation of $\pi/2$ on the unit circle. We create a uniform grid over the unit circles for $U_{\cdot 1}$ and $Y$. Fig. 2 visualizes the *severity* of non-quasiconvexity of $\mathcal{L}$ as we move over this grid for three different settings of the singular values. To define the severity of non-quasiconvexity, let $\lambda_{\text{worst}}$, $\lambda^*$, and $\lambda_{\text{worst-min}}$ correspond to the $\lambda$ maximizing $\mathcal{L}$, the $\lambda$ minimizing $\mathcal{L}$, and the $\lambda$ corresponding to the local minimum with largest $\mathcal{L}$, respectively. We then compute

$$\text{severity} := \frac{\mathcal{L}(\lambda_{\text{worst-min}}) - \mathcal{L}(\lambda^*)}{\mathcal{L}(\lambda_{\text{worst}}) - \mathcal{L}(\lambda^*)} \tag{3}$$

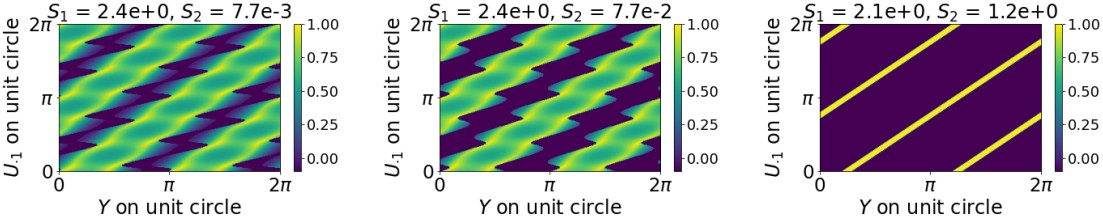

Figure 2: How does quasiconvexity depend on $Y$ and $U$, for data with $N = 3$ and $D = 2$? The left, center, and right panels each correspond to a different setting of the singular values $S$. We divide the unit circle for each of $U$ and $Y$ into 225 equally spaced points. We examine the severity of non-quasiconvexity in $\mathcal{L}$ over this $225 \times 225$ grid. Each grid point is colored by Eq. (3), where dark purple corresponds to quasiconvexity (no local minima).

If the severity is near 1, then finding the worst local minimum is nearly as bad as selecting the worst possible $\lambda$. In Fig. 2 we color pixels according to this measure of severity and do in fact see values near 1. We see that, in general, $U$, $Y$, and $S$ have a complicated interaction to determine the quasiconvexity of $\mathcal{L}$. The values of $Y$ that produce quasiconvexity depend on $U$. No setting of $Y$ or $U$ guarantees quasiconvexity.

Fig. 2 demonstrates the poor performance of a hypothetical optimizer that always finds the worst local minimum. In **??**, we show that poor performance is not just hypothetical; in particular, the most commonly used methods for optimizing $\mathcal{L}$ – grid search and gradient descent – encounter difficulties due to non-quasiconvexity. Issues due to non-quasiconvexity are also not specific to measuring performance in terms of $\mathcal{L}$ nor to optimizing the LOOCV loss. In **??**, we show that the $K$-fold CV loss also suffers from non-quasiconvexity and that computing Eq. (3) with the test loss instead of $\mathcal{L}$ also exhibits issues due to non-quasiconvexity.

Despite the complexity of Fig. 2, one trend does seem clear: as the singular values become more similar (moving from left to right in the panels of Fig. 2), the fraction of $Y$ values and $U$ values that correspond to quasiconvexity (dark purple regions) grows. Based on this behavior, one might conjecture that a sufficiently uniform spectrum of the covariate matrix could guarantee the quasiconvexity of $\mathcal{L}$.

## 5  Quasiconvexity of $\mathcal{L}$ with a nearly uniform spectrum $S$

We now build on the conjecture of the previous section to show that we can, in fact, guarantee quasiconvexity in certain cases. In particular, we will show conditions under which a sufficiently uniform spectrum $S$ of the covariate matrix $X$ guarantees that $\mathcal{L}$ is quasiconvex.

One might hope that for large $N$, eventually any $Y$ or $U$ would yield quasiconvexity. However, even when $S$ is exactly uniform, our experiments in Section 6 show that we cannot expect such a statement. Rather, we will devise conditions on $Y$ and $U$ to avoid "extreme" settings for either quantity. With these conditions, our main theorem will show that a sufficiently flat spectrum $S$ does indeed guarantee quasiconvexity of $\mathcal{L}$. When our theorem applies, we can safely terminate an optimization procedure at the first local minimum of $\mathcal{L}$ that we discover.

We first establish notation and then state our assumptions.

**Definition 2.** *Let $\hat{\theta} := \left(X^T X\right)^{-1} X^T Y$ be the least-squares estimate. Define the least-squares residuals $\hat{E} := Y - X\hat{\theta}$. Let $\hat{\varepsilon}_n$ be the $n$th entry of $\hat{E}$.*

Note that $\hat{\theta}$ is well-defined since we have assumed that $D < N$ and that $X$ is full-rank. For the tractability of our theory, all of our assumptions and conclusions will be asymptotic in $N$; in particular, our assumptions will use big-O and little-o statements, which are to be taken with respect to $N$ growing large. In our discussion of these assumptions, we assume that $D$ is fixed. There is nothing in our proofs or assumptions that requires $D$ to be fixed; however the validity of our assumptions is not clear if $D$ grows with $N$, so we do not consider this case here. Since LOOCV is useful precisely for

finite $N$, we are careful to show in our experiments (Section 6) that these asymptotics take hold for small $N$.

Our first assumption concerns the magnitude of the residuals $\hat{E}$.

**Assumption 1.** $(1/N) \sum_{n=1}^{N} \hat{\varepsilon}_n^2$ *is* $O(1)$ *(i.e. it does not grow with $N$).*

This assumption is fairly lax. For example, suppose our linear model is well-specified. In particular, suppose there exists some $\theta^* \in \mathbb{R}^D$ such that $y_n = \langle x_n, \theta^* \rangle + \varepsilon_n$ where the $\varepsilon_n$ are i.i.d. $\mathcal{N}(0, \sigma^2)$ for some $\sigma > 0$. Stack the $\varepsilon_n$ into a vector $E \in \mathbb{R}^N$. Then $\|\hat{E}\|^2 = \|(I_N - UU^T)E\|^2 < \|E\|^2$. Since $(1/N)\|E\|^2$ is $O(1)$ with high probability, it follows that Assumption 1 holds with high probability in this well-specified linear model. We emphasize that Assumption 1 depends on the residuals of the least squares estimate, not (directly) on the noise in the observations.

Our next assumption governs the size of the least squares estimate $\hat{\theta}$.

**Assumption 2.** $\|\hat{\theta}\|$ *is* $O(1)$ *(i.e. it does not grow with $N$).*

Again, this is a lax assumption. For example, given any statistical model for which $\hat{\theta}$ is a consistent estimator for some quantity, Assumption 2 holds.

Our next assumption constrains the uniformity of the left-singular value matrix $U$ with rows $u_n \in \mathbb{R}^D$.

**Assumption 3.** *We have* $\max_n \|u_n\|^2 := \|u_{\max}\|^2 = O(N^{-p})$ *for some $p > 1/2$.*

Assumption 3 is an assumption about the coherence of the $U$ matrix, a quantity of importance in compressed sensing and matrix completion [Candés and Recht, 2009]. In particular, Assumption 3 requires that the coherence of $U$ decay sufficiently fast as a function of $N$. Suppose we remove the condition that $U$ have zero-mean columns (see Condition 1 and the discussion after Remark 1) and assume a uniform distribution over valid $U$ (i.e. matrices with orthonormal columns); then Assumption 3 is known to hold with high probability for any $p \in (1/2, 1)$ [Candés and Recht, 2009, Lemma 2.2].

There do exist matrices $U$ with orthonormal zero-mean columns that do not satisfy Assumption 3. For instance, take some small $N_0$ (say $N_0 = 5$) and a valid $U'$ for this $N_0$. Then, for $N > N_0$, form $U$ by appending $\mathbf{0} \in \mathbb{R}^{(N-N_0) \times D}$ to the bottom of $U'$. This construction yields an $N \times D$ matrix $U$ with orthonormal and zero-mean columns for which $\|u_{\max}\|^2$ is constant as $N$ grows. Still, in our experiments in Section 6 and **??**, we confirm that, for a uniform distribution over orthonormal $U$ with zero-mean columns, Assumption 3 holds with high probability.

Our final assumption is a technical assumption relating $\|u_n\|^2$, $\hat{\varepsilon}_n$, and $\hat{\theta}$.

**Assumption 4.** *The following quantity is positive and $\Theta(1)$ (i.e. is bounded away from zero and does not grow with $N$):* $\|\hat{\theta}\|^2 - \sum_{n=1}^{N} \|u_n\|^2 \left( \langle u_n, \hat{\theta} \rangle^2 + 2\hat{\varepsilon}_n^2 \right)$

Roughly, this assumptions means that the largest $\|u_n\|^2$ and $\hat{\varepsilon}_n^2$ values do not occur for the same values of $n$. To see this relation, note that Assumption 3 implies $\|\hat{\theta}\|^2 - \sum_n \|u_n\|^2 \langle u_n, \hat{\theta} \rangle^2 \geq (1 - O(N^{-p}))\|\hat{\theta}\|^2$. If we assume that $\|\hat{\theta}\|^2 = \Theta(1)$ (i.e. Assumption 2 holds and $\hat{\theta}$ does not converge to $\mathbf{0} \in \mathbb{R}^D$), then we find $\|\hat{\theta}\|^2 - \sum_n \|u_n\|^2 \langle u_n, \hat{\theta} \rangle^2 = \Theta(1)$. So, we need only that $\sum_n \|u_n\|^2 \hat{\varepsilon}_n^2 = o(1)$ for Assumption 4 to hold; e.g. we need that the largest values of $\|u_n\|^2$ and the largest values of $\hat{\varepsilon}_n^2$ typically do not occur for the same values of $n$.

With our assumptions in hand, we can now state our main theorem. Our theorem relates the uniformity of the spectrum of $X$ to the quasiconvexity of $\mathcal{L}$. As we have shown in Proposition 2, the scaling of the singular values does not matter for the quasiconvexity of $\mathcal{L}$. We therefore take the spectrum to be nearly uniform around $\mathbf{1} \in \mathbb{R}^D$.

**Theorem 1.** *Consider a series of regression problems with $N$ growing to infinity, where the $N$th problem uses data $(X^{(N)}, Y^{(N)})$. Assume this sequence satisfies Assumptions 1 to 4. Let the covariate matrix of the $N$th regression problem have SVD $X^{(N)} = U^{(N)} \operatorname{diag}(S^{(N)}) V^{(N)}$. There is a $N_0 > 0$ and neighborhood $\Delta$ of $\mathbf{1} \in \mathbb{R}^D$ such that if $N \geq N_0$ and the spectrum $S^{(N)} \in \Delta$, then $\mathcal{L}$ is quasiconvex.*

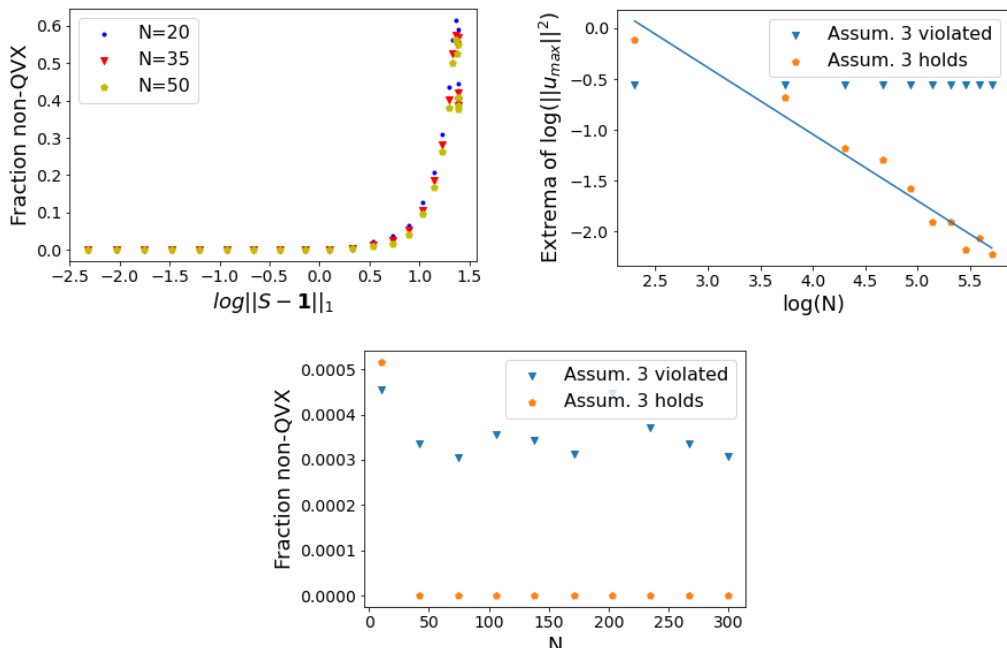

Figure 3: (*Upper left*): We generate many datasets and plot the fraction that are non-quasiconvex ("non-QVX"), varying $N$ and the distance of the spectrum from uniformity ($\|S - \mathbf{1}\|_1$). (*Upper right*): We generate two sets (orange, blue) of left-singular vector matrices $U$. In the blue case, we check that the maximum of $\log \|u_{\max}\|^2$ across all $U$ for a particular $N$ decreases roughly linearly on a log-log plot (i.e. the blue set satisfies Assumption 3). In the orange case, we check that the minimum of $\log \|u_{\max}\|^2$ across all $U$ for a particular $N$ is roughly constant (i.e. the orange set does not satisfy Assumption 3). (*Lower*): For all the $U$ matrices from the upper right plot, we generate many datasets and plot the fraction that are not quasiconvex.

*Proof sketch:* For one-dimensional functions $\mathcal{L}$, a sufficient condition for quasiconvexity is that for all $\lambda$ such that $\mathcal{L}'(\lambda) = 0$, we have $\mathcal{L}''(\lambda) > 0$ [Boyd and Vandenberghe, 2009, Chapter 3.4]. We first show $\mathcal{L}'$ can be zero only for a bounded set of $\lambda$. We then show that for any $\lambda$ within this set with $\mathcal{L}'(\lambda) = 0$, we have $\mathcal{L}''(\lambda) > 0$. See **??** for a full proof. $\qquad\square$

In Section 4, we showed it can be difficult to guess when $\mathcal{L}$ is quasiconvex. But Theorem 1 yields one condition that guarantees $\mathcal{L}$ is quasiconvex: when $X$ has a nearly uniform spectrum. A natural question then is: when is the spectrum of $X$ nearly uniform? As it happens, a uniform spectrum occurs under standard assumptions, for example, when the $x_{nd}$ are i.i.d. sub-Gaussian random variables.

**Definition 3** (e.g. [Vershynin, 2018]). *A random variable $Q$ is sub-Gaussian if there exists a constant $c > 0$ such that $\mathbb{E}[\exp(Q^2/c^2)] \leq 2$.*

**Corollary 1.** *Take any series of regression problems satisfying Assumptions 3 and 4. Assume the series of regression problems are drawn from a well-specified linear model for some $\theta^* \in \mathbb{R}^D$: $y_n^{(N)} = \langle x_n^{(N)}, \theta^* \rangle + \varepsilon_n$, where $\varepsilon_n \overset{i.i.d.}{\sim} \mathcal{N}(0, \sigma^2)$. If $\sigma$ is sufficiently small, $\hat{\theta}$ is consistent for $\theta^*$, and the entries of the covariate matrices $x_{nd}^{(N)}$ are i.i.d. sub-Gaussian random variables, then $\mathcal{L}$ is quasiconvex with probability tending to 1 as $N \to \infty$.*

*Proof sketch:* Assumptions 1 and 2 hold for a well-specified linear model. If the entries of $X$ are i.i.d. sub-Gaussian random variables, standard concentration inequalities imply that its spectrum is nearly uniform with high probability; hence the result of Theorem 1 applies. See **??** for a full proof. $\qquad\square$

# 6 Theorem 1 in practice

In Section 5, we established a number of assumptions that we then required in Theorem 1 to prove that $\mathcal{L}$ is quasiconvex. A few questions remain about our theorem in practice: (1) how large is the neighborhood $\Delta$, (2) how necessary are our assumptions, and (3) how large do we require $N$ to be? We explicitly answer (1) and (2) with experiments below. (3) is particularly concerning, as regularization has minimal impact when $N \gg D$. That is, there is little performance gain by using a regularizer, which removes the need for hyperparameter tuning. To show that our theorem holds when $N \sim D$, the majority of our experiments validating Theorem 1 use $N$ that is at most an order of magnitude larger than $D$.

Throughout our experiments, we check for non-quasiconvexity numerically and use a shortcut formula to compute $\mathcal{L}$ that takes advantage of the fact that the right singular vectors of $X$ are $V = I_D$; see **??** for details. The only software dependency for our experiments is NumPy [Harris et al., 2020], which uses the BSD 3-Clause "New" or "Revised" License.

**How do we know S is in the neighborhood $\Delta$ in Theorem 1?** While our theorem does not give an explicit size of the neighborhood $\Delta$, we can show empirically that $\Delta$ is substantial, even for small to moderate $N$. We fix $D = 5$. To generate various spectra of $X$, we set $S_d = e^{\alpha d}/e^{\alpha D}$. For $\alpha \to 0$, we get $S \to \mathbf{1}$; we vary $\alpha$ from zero to one to generate spectra of varying distances from uniformity. For each $\alpha$, we sample 100 left-singular-value matrices $U$ from the uniform distribution over orthonormal $U$ with column means equal to 0; see **??** for how to generate such matrices. We fix a unit-norm $\theta^* \in \mathbb{R}^D$ and for each $U$, we generate data from a well-specified linear model, $y_n = \langle x_n, \theta^* \rangle + \varepsilon_n$, where the $\varepsilon_n$ are drawn i.i.d. from $\mathcal{N}(0, \sigma^2)$ with variance $\sigma^2 = 0.5$. In particular, for each setting of $U$, we generate 100 vectors $Y$. For each setting of $U$ and $Y$, we compute $\mathcal{L}$ and check whether it is quasiconvex. In the top left panel of Fig. 3, we report the fraction of problems (out of the $100 * 100 = 10,000$ datasets for the corresponding $\alpha$ value) with a non-quasiconvex $\mathcal{L}$ versus the distance from uniformity, $\|S - \mathbf{1}\|_1$. We see that, even for $N = 20$, the fraction of non-quasiconvex problems quickly hits zero as $\|S - \mathbf{1}\|_1$ shrinks.

To provide a rough practical heuristic, we observe from Fig. 3 that when $N = 50$ and $\|S - \mathbf{1}\|_1 \sim 2$, non-quasiconvexity occurs less than 1% of the time. In Fig. 3, $\Delta$ appears to grow slightly with $N$, so Fig. 3 seems to suggest that we should expect to see little quasiconvexity if $\|S - \mathbf{1}\|_1 \leq 2$ and $N \geq 50$. In practice, how should we access the spectrum to check this condition? When $N$ and $D$ are small enough we can directly compute $S$ via the singular value decomposition; however, in practice, $N$ or $D$ may be large. If $N$ is large and $D$ is small, we can access the spectrum as the square root of the eigenvalues of $X^T X$. If $N$ is *very* large, formation of $X^T X$ can be expensive; in this case, we suggest the use of randomized sketching to obtain a randomized approximation to the spectrum of $X$ [Woodruff et al., 2014]. Finally, when both $N$ and $D$ are large, we can use spectral density estimation, which gives an estimate of the density of a matrix's eigenvalues [Lin et al., 2016], and has been shown to successfully scale to large problems [Ghorbani et al., 2019, Yao et al., 2020].

**Importance of Assumption 3.** We now establish the necessity of Assumption 3 on the decay of $\|u_{\max}\|^2$ with $N$. To do so, we generate two sets of matrices $U$ as $N$ grows. We generate the first set to satisfy Assumption 3, and we generate the second to violate Assumption 3. In both cases, we will take $D = 5$ and ten settings of $N$ between $N = 10$ and $N = 300$.

To generate the assumption-satisfying matrices $U$, we proceed as follows. For each $N$, we draw 500 matrices $U$ from the uniform distribution over orthonormal $U$ matrices with column means equal to 0. For each $N$, we plot the *maximum* value of $\|u_{\max}\|^2$ across these 500 $U$ matrices in Fig. 3 (top-right) as a blue dot. We fit a line to these values on a log-log plot, and find the slope is -0.74. This confirms that these matrices satisfy Assumption 3.

To generate assumption-violating matrices $U$, we proceed as follows. Recall that the smallest $N$ is 10 and $D = 5$. 100 times, we randomly draw a $U_{\text{small}} \in \mathbb{R}^{8 \times 5}$. We then construct each $U$ by appending $N - 8 \times D$ zeros to $U_{\text{small}}$. For each $N$, we plot the *minimum* value of $\|u_{\max}\|^2$ across these 100 $U$ matrices in Fig. 3 (top-right) as an orange dot. Since the minimum of $\|u_{\max}\|^2$ is constant with $N$, Assumption 3 is violated.

Now we check quasiconvexity. To that end, we randomly select a fixed unit-norm vector $\theta^* \in \mathbb{R}^D$. For each $N$, we generate 100 noise vectors $E \in \mathbb{R}^N$, where the entries $E_n$ are drawn i.i.d. from $\mathcal{N}(0, 0.5)$. For each $U$ and $E$, we construct $Y = US\theta^* + E$, where $S = \mathbf{1} \in \mathbb{R}^D$. We then compute

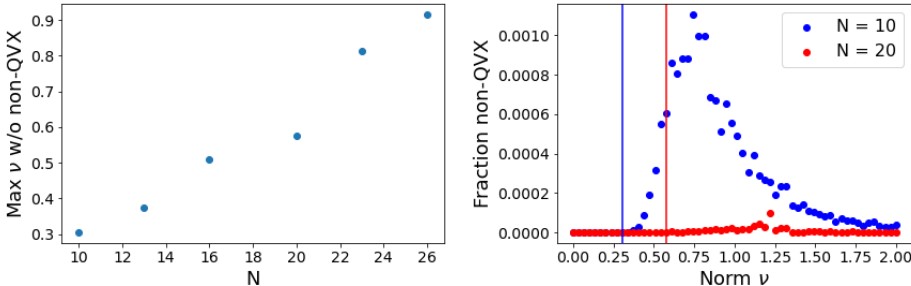

Figure 4: Checking Assumption 1. (*Left*): For each $N$ and each $\nu = \|\hat{E}\|$, we generate many datasets and check if there is any non-quasiconvex ("non-QVX") $\mathcal{L}$. We plot the largest $\nu$ for which we find only quasiconvex $\mathcal{L}$. The growth is roughly linear, which suggests Assumption 1 cannot be loosened. (*Right*): For each $N$ and each $\nu = \|\hat{E}\|$, we generate many data sets and plot the fraction of $\mathcal{L}$ that are non-quasiconvex. Vertical lines show $\nu_{max}$ for each $N$.

the fraction of these ($100 * 100 = 10{,}000$) losses $\mathcal{L}$ that are non-quasiconvex. In the lower panel of Fig. 3, we plot this fraction against $N$ for both for the assumption-satisfying case (blue) and the assumption-violating case (orange) in Assumption 3. When the assumption is satisfied (blue), we see that the conclusion of Theorem 1 holds: beyond a certain $N$, there are no settings of $U$ or $Y$ that generate a non-quasiconvex $\mathcal{L}$. We see that in practice, the boundary $N$ is small or moderate (below 50). When the assumption is violated (orange dots), we see that the conclusion of Theorem 1 fails to hold: as $N$ grows, there are still settings of $U$ and $Y$ for which quasiconvexity fails to hold. Finally, we call attention to the vertical axis. Even in the assumption-violating case, the fraction of non-quasiconvex losses is small. It follows that, even for our degenerate $U$'s, nearly every combination of noise and $U$ leads to a quasiconvex $\mathcal{L}$. An interesting challenge for future work is to provide a precise characterization of this effect.

**Do the $\hat{\varepsilon}_n$ need to be small (Assumption 1)?** Finally, we demonstrate the necessity of Assumption 1, which can be restated as requiring that $\|\hat{E}\|^2$ grows at most linearly in $N$. But we also find a suggestion that there may be even more permissive assumptions of interest. To this end, we vary $N$ from 10 to 30. For each $N$, we generate 4,000 settings of $U$, each uniform over orthonormal $U$ with column means equal to 0. For each $U$, we generate 250 unit vectors $R$ such that $U^T R = 0$ (each generated uniformly over such $R$; see **??**). Separately, we consider 60 different norms $\nu$ for the vector $\hat{E}$ equally spaced between $\nu = 0$ and $\nu = 2$; these are the same across $N$. We generate a single unit-norm $\theta^* \in \mathbb{R}^D$.

For each setting of $U$, $R$, and $\nu$, we consider the regression problem with covariate matrix $U\mathbf{1}$ and responses $Y = U\mathbf{1}\theta^* + \hat{E}$, where $\hat{E} := \nu R$. We record whether $\mathcal{L}$ is quasiconvex or not for this problem. For a particular $N$ and particular error-norm $\nu$, we check whether *any* of the $\mathcal{L}$ (across $4{,}000 * 250 = 1{,}000{,}000$ problems) were non-quasiconvex. Finally, for each $N$, we find the maximum error-norm $\nu_{\max,N}$ such that for all $\nu < \nu_{\max,N}$ every regression problem is quasiconvex. We plot a dot at $(N, \nu_{\max,N})$ in the left panel of Fig. 4. We see that in fact the boundary of allowable $\hat{E}$ norms does grow about linearly in $N$.

Our next plot lends additional insight into how the boundary $\nu_{\max,N}$ varies with $N$ and is also suggestive of other potential variations on Assumption 1 that might be of interest. In particular, in the right panel of Fig. 4, we consider two particular values of $N$: $N = 10$ (blue) and $N = 20$ (red). For each setting of $\nu$ on the horizontal axis, we compute the fraction of non-quasiconvex losses $\mathcal{L}$ over all settings of $U$ and $R$ ($4{,}000 * 250 = 1{,}000{,}000$ problems for each $\nu$). We see that, as expected from the left panel of Fig. 4, the boundary $\nu_{\max,N}$ is higher for $N = 20$ than for $N = 10$. Surprisingly, we also see that at high values of $\nu$, the fraction of non-quasiconvex cases decreases again. We conjecture that in general (i.e. beyond these two particular $N$), large amounts of noise leads to little or no non-quasiconvexity. Finally, we note that, as in the bottom panel of Fig. 3, the fraction of non-quasiconvex cases across all $\nu$ is low. Again, this small fraction suggests a direction for future work.

# 7 Discussion

We have shown that the LOOCV loss $\mathcal{L}$ for ridge regression can be non-quasiconvex in real-data problems. Local optima need not be global optima. These multiple local optima may pose a practical problem for common hyperparameter tuning methods like gradient-based optimizers, which may get stuck in a local optimum, and grid search, for which upper and lower bounds need to be set.

We proved that the quasiconvexity of $\mathcal{L}$ is determined by only a few aspects of a linear regression problem. But we also showed that the quasiconvexity of $\mathcal{L}$ is still a complicated function of the remaining quantities, and as of this writing the nature of this function is far from fully understood. Nonetheless, we have provided theory that guarantees at least some useful cases when $\mathcal{L}$ is quasiconvex: when the spectrum of the covariate matrix is sufficiently flat, the least-squares fit $\hat{\theta}$ fits the data reasonably well, and the left singular vectors of the covariate matrix are regular. In our experiments, we have confirmed that these assumptions are necessary to some extent: when they are not satisfied, $\mathcal{L}$ can be non-quasiconvex. Still, our empirical results make it clear there is more to be explored. We describe some of the directions we believe are most interesting for future work below.

**Sharper characterization of when $\mathcal{L}$ is quasiconvex.** Fig. 2 shows that non-quasiconvexity disappears as the spectrum of $X$ becomes uniform; however, it is clear that there is very regular behavior to the pattern of quasiconvexity even when the singular values of $X$ are non-uniform. We are not able to characterize these patterns at this time but believe these patterns pose a fascinating challenge for future work. Relatedly, our experiments (Section 6) show that when our assumptions are violated, quasiconvexity of $\mathcal{L}$ is not guaranteed. However, we have observed that even when $\mathcal{L}$ is not guaranteed to be quasiconvex, many settings of $U$ and $Y$ still give quasiconvexity. In many of our experiments, the fraction of non-quasiconvex losses $\mathcal{L}$ was extremely small.

**How many local optima and how bad are they?** Without the guarantee of a single, global optimum, it is not clear that we can ever know that we have globally (near-)optimized $\mathcal{L}$. However, notice that our examples in Fig. 1 all have at most two local optima. In simulated experiments, we also typically encountered two local optima in non-quasiconvex losses, although we have not studied this behavior systematically. If $\mathcal{L}$ were guaranteed to have only two or some small number of optima, optimization might again be straightforward, even in the case of non-quasiconvexity; an algorithm could search until it finds the requisite number of optima and then report the one with the smallest value of $\mathcal{L}$. Alternatively, one might hope that all local optima have CV loss (and ideally out-of-sample error) close in value to that of the global optimum. Indeed, Kawaguchi [2016] argue that this property holds for certain losses in deep learning. Presumably it is not universally the case that local optima exhibit similar loss since the right panel of Fig. 1 seems to give a counterexample. But it might be widely true, or true under mild conditions. Meanwhile, in the absence of such guarantees, optimization of $\mathcal{L}$ should proceed with caution.

**Beyond ridge regression.** We have shown – in our opinion – surprising non-quasiconvexity for the LOOCV loss for ridge regression. Do similar results hold for simple models outside ridge regression? The regularization parameter in other $\ell_2$ or $\ell_1$-regularized generalized linear models is often tuned by minimizing a cross-validation loss. In preliminary experiments, we have found non-quasiconvexity in $\ell_2$-regularized logistic regression. To what extent do empirical results like those in Fig. 2 or theoretical results like those in Theorem 1 hold for other models and regularizers?

### Acknowledgements

We thank the anonymous reviewers for suggesting valuable additional experiments and very carefully checking (and correcting) our proofs. WS and TB thank an NSF Career Award and an ONR Early Career Grant for support. MU and ZF gratefully acknowledge support from NSF Award IIS-1943131, the ONR Young Investigator Program, and the Alfred P. Sloan Foundation.

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
