## A  Real dataset descriptions

Here, we give the details for the real datasets used to generate Fig. 1.

**Life expectancy.**  Our first real dataset contains $N = 2{,}938$ observations of life expectancy in a country, along with $D = 20$ covariates such as country of origin or alcohol use. The dataset is available from [Rajarshi, 2018]. In this case, $\mathcal{L}$ for the full dataset is quasiconvex. But now consider some standard data pre-processing. Practitioners often perform principal component regression (PCR) with the aim of reducing noise in the estimated $\theta$. That is, they take the singular value decomposition of $X = USV$; they then produce an $N \times R$ dimensional covariate matrix $X'$ by retaining just the upper $R$ singular values of $X$: $X' = U_{.,:R}S_{:R}$. If we include this pre-processing step, the resulting LOOCV curve $\mathcal{L}$ is non-quasiconvex for many values of $R$; in the center panel of Fig. 1 we show one example for $R = 15$.

This dataset does contain information about people. However, it is only reported at the aggregated level by a given country per year. It is not clear to us whether or not consent was obtained by the individuals living in these countries; however, we feel the publication of such data is unlikely to negatively affect any given individual. Additionally, while we do not know if the data reveals any identifying information about an individual, we feel it is unlikely to do so, as it is published at the country level.

**Wine dataset.**  Our second dataset consists of recorded wine quality of $N = 1{,}599$ red wines. The goal is to predict wine quality from $D = 11$ observed covariates relating to the chemical properties of each wine [Cortez et al., 2009a,b]. We find that subsets of this dataset often exhibit non-quasiconvex $\mathcal{L}$. We search over 400 random subsets of this dataset of size $N = 50$. In Fig. 1. Twelve of these led to non-quasiconvex losses $\mathcal{L}$, and Fig. 1 shows one of these examples.

This dataset does not contain information about people, and so concerns about consent and personally identifying information do not seem relevant here.

## B  Proof of Proposition 2

We now restate and then prove Proposition 2.

**Proposition 2.** *Assume Condition 1 holds. The quasiconvexity of $\mathcal{L}$ is independent of the following*

1. *The matrix of right singular vectors, $V$*

2. *The norm of the responses, $\|Y\|_2$*

3. *The scaling of the singular values (i.e. changing $S$ into $S/c$ for $c \in \mathbb{R}_{>0}$)*

*in the sense that altering any of these quantities does not change whether or not $\mathcal{L}$ is quasiconvex.*

*Proof.*  First, it is easiest to write our function of interest in a simpler form:

$$\mathcal{L}(\lambda) = \sum_{n=1}^{N} \frac{1}{(1 - Q_n(\lambda))^2} (x_n^T \hat{\theta}_\lambda - y_n)^2, \tag{4}$$

where $Q_n(\lambda) := x_n^T \left( X^T X + \lambda I_D \right)^{-1} x_n$ and $\hat{\theta}_\lambda := \left( X^T X + \lambda I_D \right)^{-1} X^T Y$.

Let the singular value decomposition of $X$ be $X = U \operatorname{diag}(S) V$. Then:

**$V$ does not affect does not affect the quasiconvexity of $\mathcal{L}$.**  To prove this claim, note that $x_n = u_n^T \operatorname{diag}(S) V$, where $u_n$ is the $n$th row of $U$. So:

$Q_n(\lambda) = u_n^T \operatorname{diag}(S) V^T \left( V \operatorname{diag}(S^2 + \lambda)^{-1} V^T \right) V \operatorname{diag}(S) u_n = u_n^T \operatorname{diag}(S) \operatorname{diag}(S^2 + \lambda)^{-1} \operatorname{diag}(S) u_n$.

So $Q_n(\lambda)$ is actually independent of $V$. Next,

$$x_n^T \hat{\theta}_\lambda = u_n^T \operatorname{diag}(S) V^T V \operatorname{diag}(S^2 + \lambda)^{-1} V^T V \operatorname{diag}(S) U^T Y$$
$$= u_n^T \operatorname{diag}(S) \operatorname{diag}(S^2 + \lambda)^{-1} \operatorname{diag}(S) U^T Y,$$

which is also independent of $V$.

$\|\mathbf{Y}\|_2^2$ **does not affect the quasiconvexity of** $\mathcal{L}$. In particular, we can treat $Y$ as sitting on the $D$-dimensional unit sphere. To see this, take two different $Y$'s related by a scaling: $y_n^{(1)} = c y_n^{(0)}$ for some scalar $c \in \mathbb{R}$. Then, using the same superscripts:

$$\hat{\theta}_\lambda^{(1)} = \left( X^T X + \lambda I_D \right)^{-1} X^T Y^{(1)} = c \hat{\theta}_\lambda^{(0)}.$$

So, we can relate the two LOOCV functions by:

$$\mathcal{L}^{(1)}(\lambda) = \sum_n \frac{1}{(1 - Q_n(\lambda))^2} \left( c x_n^T \hat{\theta}_\lambda^{(0)} - c y_n^{(0)} \right)^2 = c^2 \mathcal{L}^{(0)}(\lambda). \tag{5}$$

So scaling $Y$ by $c$ uniformly scales $\mathcal{L}(\lambda)$ by $c^2$. Mutliplying $\mathcal{L}$ by a constant does not affect is quasiconvexity.

**The scaling of the singular values** $s_1, \ldots, s_D$ **does not affect the quasiconvexity of** $\mathcal{L}$. We have already shown that $V$ does not affect the quasiconvexity of $\mathcal{L}$, so fix $V = I_D$ to simplify the proof. Pick some scaling $c > 0$, and fix some spectrum $S^{(1)}$. Define $S^{(0)} := c S^{(1)}$. Using the same superscripts, we have:

$$Q_n^{(0)}(\lambda) = \sum_{d=1}^D u_{nd}^2 \frac{c^2 s_d^2}{c^2 s_d^2 + \lambda} = \sum_{d=1}^D u_{nd}^2 \frac{s_d^2}{s_d^2 + (\lambda/c^2)} = Q^{(1)} \left( \frac{\lambda}{c^2} \right) \tag{6}$$

Similarly, define $(x_n^T \hat{\theta})^{(0)}(\lambda)$ to be the inner product of $x_n^{(0)}$ and $\hat{\theta}^{(0)}(\lambda)$. Then:

$$(x_n^T \hat{\theta})^{(0)}(\lambda) = u_n^T \operatorname{diag}(S^{(0)}) \left( \operatorname{diag}(S^{(0)})^2 + \lambda I_D \right)^{-1} \operatorname{diag}(S^{(0)}) U^T Y \tag{7}$$

$$= u_n^T \operatorname{diag}(S^{(1)}) \left( \operatorname{diag}(S^{(1)})^2 + \frac{\lambda}{c^2} I_D \right)^{-1} \operatorname{diag}(S^{(1)}) U^T Y \tag{8}$$

$$= (x_n^T \hat{\theta})^{(1)} \left( \frac{\lambda}{c^2} \right). \tag{9}$$

This, along with Eq. (6) implies that $\mathcal{L}^{(0)}(\lambda) = \mathcal{L}^{(1)}(\lambda/c^2)$. That is, multiplying the singular values by $c$ stretches out $\mathcal{L}$ by a factor of $c$. This does not change the quasiconvexity of $\mathcal{L}$. $\qquad \square$

## C Measuring severity of non-quasiconvexity with other losses and optimization methods

Fig. 2 shows the severity of non-quasiconvexity when searching over all possible $N = 3, D = 2$ regression problems for a fixed spectrum $S$. Recall that we measured the *severity* of non-quasiconvexity as

$$\text{severity} := \frac{\mathcal{L}(\lambda_{\text{worst-min}}) - \mathcal{L}(\lambda^*)}{\mathcal{L}(\lambda_{\text{worst}}) - \mathcal{L}(\lambda^*)}, \tag{10}$$

where $\lambda_{\text{worst}}$, $\lambda^*$, and $\lambda_{\text{worst-min}}$ are the $\lambda$ maximizing $\mathcal{L}$, the $\lambda$ minimizing $\mathcal{L}$, and the $\lambda$ corresponding to the local minimum with largest $\mathcal{L}$, respectively. Eq. (10) measures the relative quality of the optima found by a hypothetical optimizer that always finds the worst possible local optimum, where the quality of an optimum is measured by $\mathcal{L}$. We view even modest values of this measure of severity as fairly severe: e.g. a severity of 0.1 indicates that the excess loss incurred by finding a bad minimum is 10% of the excess loss incurred by using the worst possible $\lambda$. Here, we investigate what happens when we replace different parts of Eq. (10).

**Realistic optimization methods.** First, we ask what would happen when using more realistic optimization methods. In particular, we consider replacing $\lambda_{\text{worst-min}}$ by the $\lambda$ of the minimum reported by either gradient descent or grid search. For gradient descent, we have to choose a $\lambda$ at which to initialize. There does not seem to be any natural data-driven heuristic to initialize gradient descent with, so we always initialize at $\lambda = 0$. For grid search, we need to choose the range of our grid and how fine a grid we will use. For the leftmost point of our grid, we use $\lambda = 0$. For the rightmost point of our grid, we use $\lambda = 2$. To justify this choice of a maximum $\lambda$, recall we have set

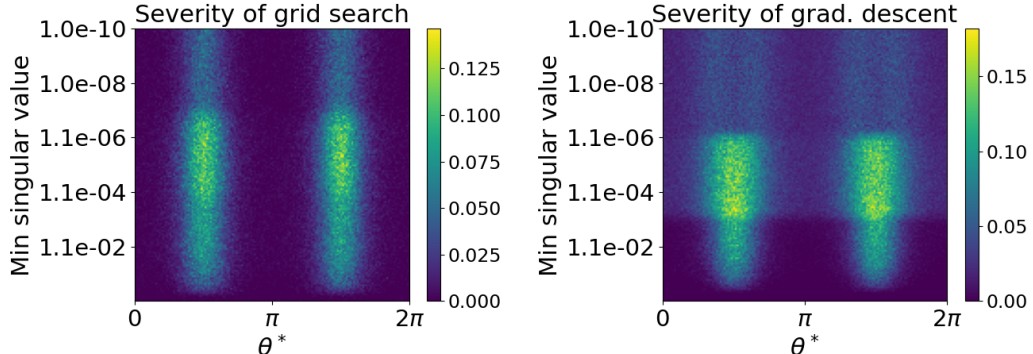

Figure 5: (*Left*): Severity of non-quasiconvexity where $\lambda_{\text{worst-min}}$ is the $\lambda$ recovered by grid search. (*Right*): Severity of non-quasiconvexity where $\lambda_{\text{worst-min}}$ is the $\lambda$ recovered by gradient descent. Note the color scales on the left and right figures differ slightly and also differ significantly from Fig. 2 in the main text.

the maximum singular value to be equal to 1. Thus $\lambda > 1$ implies that the regularizer has exceeded the scale of the covariates; it seems reasonable to not consider $\lambda$ too much larger than this scale, so we select $\lambda = 2$ as our maximum. We assume that if no minimum is encountered in the range of our grid that any reasonable user would continue the grid search until finding a minimum; in such a case, we say that grid search reports the minimum with the smallest $\lambda$ value. Given this range of $\lambda$'s, we set our grid to be 200 $\lambda$'s evenly spaced grid on the log-scale. We have never found a difference in our experiments by making this grid finer, so we expect this has no effect on the results presented here.

We could replicate the $N = 3, D = 2$ experiment of Fig. 2 using grid search and gradient descent. However, it seems that in the case of $N = 3, D = 2$, the leftmost minimum of $\mathcal{L}$ is always at some $\lambda < 1$ and is always the global optimum of $\mathcal{L}$. So, our implementations of grid search and gradient descent always find the global optimum of $\mathcal{L}$ in this case. To create examples with more interesting behavior, we instead consider the case of $N = 50, D = 2$. Here, we can still parameterize $\theta^*$ by a scalar on the unit circle, but cannot do the same for $U$. Instead, we vary $\theta^*$ and the singular values of $X$ over a grid and average the severity over many random setings of $U$ for each setting of $S$ and $\theta^*$. We set the singular values of $X$ to be $S_1 = 1, S_2 = \alpha$, for some $\alpha \in (0, 1]$. We let $\alpha$ vary on a log-scale grid between 0 and 1 and $\theta^*$ on a grid between 0 and $2\pi$. For each setting of $\alpha$ and $\theta^*$, we construct 1000 random linear regression problems by drawing $U$ at uniform from all zero-column-mean orthonormal matrices (Appendix G). We then set $Y = U \operatorname{diag}(S)\theta^* + E$, where $E \in \mathbb{R}^N$ has i.i.d. $\mathcal{N}(0, 0.1)$ entries. We run either grid search or gradient descent on the resulting LOOCV loss $\mathcal{L}$ and compute the severity of non-quasiconvexity via Eq. (10) with $\lambda_{\text{worst-min}}$ replaced by the $\lambda$ returned by grid search or gradient descent. Finally, we average the severity over all 1000 trials. Fig. 5 visualizes the results for grid search (left) and gradient descent (right). We find that there exist settings of $\alpha$ and $\theta^*$ for which grid search and gradient descent typically find poor local optima. These difficulties occur for moderate values of $\alpha$ and when $\theta^*$ sits at an angle of around $\pi/2$ or $3\pi/2$. That is, when $\| \operatorname{diag}(S)\theta^* \| \approx \alpha$.

**Other CV losses.** One might ask if our finding of non-quasiconvexity is limited to leave-one-out CV. To show this is not the case, we consider the same $N = 50, D = 2$ setting as above, and again vary $\theta^*$ and $\alpha$ on a grid. We let $\mathcal{K}(\lambda)$ be the $K$-fold CV loss and compute the severity of its non-quasiconvexity as:

$$\frac{\mathcal{K}(\lambda_{\text{worst-min}}) - \mathcal{K}(\lambda^*)}{\mathcal{K}(\lambda_{\text{worst}}) - \mathcal{K}(\lambda^*)},$$

where $\lambda_{\text{worst-min}}, \lambda^*$, and $\lambda_{\text{worst}}$ are the $\lambda$ of the local minimum of $\mathcal{K}$ with highest loss, the global minimum of $\mathcal{K}$, and the $\lambda$ maximizing $\mathcal{K}$, respectively. In the left and center of Fig. 6, we show the result for 5-fold and 10-fold CV, respectively. We find similar results as before; there are settings of $\theta^*$ and $\alpha$ for which the $K$-fold CV loss has severe non-quasiconvexity.

**Test loss.** Ultimately, we are hoping to find a $\lambda$ such that the test loss $\mathcal{T}(\lambda)$ is small; we hope that minimizing $\mathcal{L}$ will give us such a $\lambda$. Here, we ask whether or not the presence of non-quasiconvexity

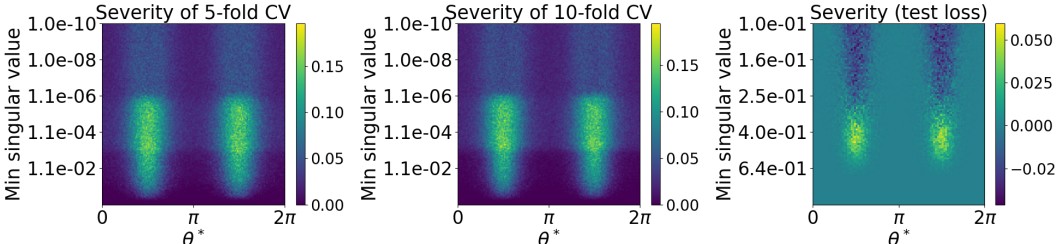

Figure 6: (*Left*): Severity of non-quasiconvexity in 5-fold CV. (*Centre*): Severity of non-quasiconvexity in 10-fold CV. (*Right*): Severity of non-quasiconvexity in $\mathcal{L}$, where the quality of $\lambda_{\text{worst}}$, $\lambda^*$ and $\lambda_{\text{worst-min}}$ are measured using the test loss. Note that the color scale on this plot is different from that on previous plots.

in $\mathcal{L}$ can make it hard to find such a $\lambda$. Here, we define the severity of non-quasiconvexity as:

$$\frac{\mathcal{T}(\lambda_{\text{worst-min}}) - \mathcal{T}(\lambda^*)}{\mathcal{T}(\lambda_{\text{worst}}) - \mathcal{T}(\lambda^*)},$$

where $\lambda_{\text{worst-min}}$, $\lambda^*$, and $\lambda_{\text{worst}}$ are the $\lambda$ of the local minimum of $\mathcal{L}$ with highest loss, the global minimum of $\mathcal{L}$, and the $\lambda$ maximizing $\mathcal{L}$, respectively. We use the same setup as above with $N = 50, D = 2$. The right of Fig. 6 shows the results. We see that there are settings of $\theta^*$ and $\alpha$ for which selecting the worst minimum of $\mathcal{L}$ typically leads to a worse test loss than does the global minimum of $\mathcal{L}$. Interestingly, there are also settings of $\theta^*$ and $\alpha$ for which using the worst minimum of $\mathcal{L}$ leads to a *better* test loss than the global minimum of $\mathcal{L}$ (negative values in the right of Fig. 6). We note that the absolute scale of severity under the test loss is smaller than in the other plots presented here, all of which measure the quality of a $\lambda$ using CV loss. This leaves open the possibility that while different local optima may have substantively different CV losses, their performance in practice – as measured by test loss – may be fairly similar.

## D  Empirical validation of Assumption 3

As noted in the main text, Assumption 3 can be interpreted as an assumption about the coherence of the $U$ matrix, a quantity commonly found in the compressed sensing literature [Candés and Recht, 2009]. In particular, Assumption 3 requires that the coherence of $U$ decay with $N$ sufficiently fast. Similar conditions have been studied in the literature for other matrices. For example, Lemma 2.2 of Candés and Recht [2009] shows that if $U$ is drawn uniformly at random from the set of all orthonormal $N \times D$ matrices, then $\max_n \|u_{\text{max}}\|^2 = O(\log(N)/N)$ with probability going to 1 as $N \to \infty$. As $\log(N)/N$ tends towards zero faster than $N^{-p}$ for any $0 < p < 1$, Lemma 2.2 of Candés and Recht [2009] proves that Assumption 3 holds with high probability if $U$ is drawn uniformly from the set of orthogonal matrices.

However, the $U$'s of interest here have an additional constraint: that their columns be zero-mean. It is not clear how to adapt the proof of Candés and Recht [2009] to this situation. Instead, we offer empirical evidence that Assumption 3 holds in the case that $U$ is drawn uniformly at random from the set of orthonormal zero-column-mean matrices. We describe how to generate such matrices in Appendix G. For fifty values of $N$ from $N = 2,500$ to $N = 20,500$, we draw 750 orthonormal zero-mean orthonormal matrices $U$ from the uniform distribution. We plot the average $\|u_{\text{max}}\|^2$ over these 750 replicas on a log scale versus $N$ in Fig. 7 (orange dots). For comparison, we plot the average $\|u_{\text{max}}\|^2$ over 750 replicas when $U$ is drawn uniformly from the set of all orthonormal matrices (no zero-mean constraint) as the blue dots. The decay of $\|u_{\text{max}}\|^2$ with and without the zero-mean constraint is essentially identical. Given this experiment, we argue that, although theoretically unjustified, Assumption 3 places only modest restrictions on the regression problems to which Theorem 1 applies.

## E  Proof of Theorem 1

We restate and prove Theorem 1 from Section 5.

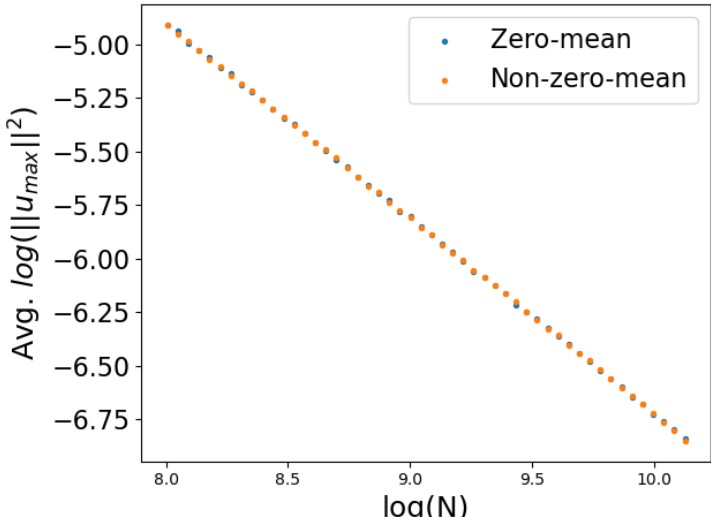

Figure 7: Experiment from Appendix D. Orange dots show the decay of $\|u_{\max}\|^2$ for uniformly drawn non-zero-column-mean orthonormal matrices $U$; we see that, as proven by Candés and Recht [2009], these matrices satisfy Assumption 3. Blue dots show the decay of $\|u_{\max}\|^2$ for uniformly drawn zero-column-mean orthonormal matrices. While such matrices are not known to satisfy Assumption 3, we see that their $\|u_{\max}\|^2$ decays at exactly the same rate as the non-zero-column-mean matrices.

**Theorem 1.** *Take any series of regression problems $\{X^{(N)}, Y^{(N)}\}_{N=1}^{\infty}$ satisfying Assumptions 1 to 4. Let the covariate matrix of the $N$th regression problem have SVD $X^{(N)} = U^{(N)} \operatorname{diag}(S^{(N)}) V^{(N)}$. There is a $N_0 > 0$ and neighborhood $\Delta$ of $\mathbf{1} \in \mathbb{R}^D$ such that if $N \geq N_0$ and the spectrum $S^{(N)} \in \Delta$, then $\mathcal{L}$ is quasiconvex.*

*Proof.* We first prove the theorem for an exactly uniform spectrum, $S = \mathbf{1}$. To do so, we work with a sufficient condition for a one-dimensional function $\mathcal{L}$ to be quasiconvex: for all $\lambda$ such that $\mathcal{L}'(\lambda) = 0$, we have $\mathcal{L}''(\lambda) > 0$ [Boyd and Vandenberghe, 2009, Chapter 3.4]. With this characterization of quasiconvexity in mind, our proof can be broken into two steps. We sketch each step here and refer to later lemmas for their proofs.

1. **Bound the region where $\mathcal{L}'$ can be zero.** Write $\mathcal{L}'$ as:

$$\mathcal{L}'(\lambda) = \frac{1}{(\lambda+1)^4} \sum_{n=1}^{N} \frac{(\lambda+1)^3}{(\lambda+1-\|u_n\|^2)^3} \left( \xi_{n1}\lambda^2 + \xi_{n2}\lambda + \xi_{n3} \right).$$

   To find where this can be zero, we can ignore the $1/(\lambda+1)^4$. Then this is *almost* a quadratic in $\lambda$, $\sum_n \xi_{n1}\lambda^2 + \xi_{n2}\lambda + \xi_{n3}$. Find the most positive root of this quadratic, $\lambda_Q = O(1)$. Bound the deviations of $\mathcal{L}'$ away from a quadratic, and bound how far these deviations can increase the zeros $\mathcal{L}'$ beyond $\lambda_Q$. **Result:** $\mathcal{L}'(\lambda)$ can only be zero for $\lambda \in [0, \lambda_Q + o(1)]$. We prove this step in Lemma 1 below.

2. **Show $\mathcal{L}''(\lambda) > 0$ for any $\lambda \in [0, \lambda_{\mathbf{Q}} + \Theta(1)]$ for which $\mathcal{L}'(\lambda) = 0$.** Essentially the same strategy; for any $]lambda$ for which $\mathcal{L}'(\lambda) = 0$, write:

$$\mathcal{L}''(\lambda) = \frac{1}{(\lambda+1)^5} \sum_{n=1}^{N} \frac{(\lambda+1)^4}{(\lambda+1-\|u_n\|^2)^4} \left( a_n\lambda^2 + b_n\lambda + c_n \right)$$

   This is a roughly a bowl-down quadratic with only one root bigger than zero; i.e. it is positive for $\lambda \in [0, \lambda'_Q]$, where $\lambda'_Q = \lambda_Q + \Theta(1)$ is the location of the quadratic's rightmost root. Show that the deviations away from quadratic imply that $\mathcal{L}''$ is positive for $\lambda \in [0, \lambda'_Q - o(1)] = [0, \lambda_Q + \Theta(1) - o(1)]$. We prove this step in Lemma 2 below.

With the theorem proved for an exactly uniform spectrum, we note that $\mathcal{L}'$ and $\mathcal{L}''$ are continuous functions of the spectrum $S$. As $\mathcal{L}''$ is stricly bounded away from zero on a region that contains $[0, \lambda_Q + \Theta(1)]$, by continuity in the singular values, there is a neighborhood $\Delta$ of $\mathbf{1} \in \mathbb{R}^D$ such that if $S \in \Delta$, $\mathcal{L}''(\lambda) > 0$ for all $\lambda$ for which $\mathcal{L}'(\lambda) = 0$. $\qquad\square$

Before getting into the proofs of our main lemmas, we can first rearrange $\mathcal{L}'$ into a convenient form. In Eq. (64) of Appendix I we gave a convenient form of $\mathcal{L}$ when the matrix of right singular vectors satisfies $V = I_D$. Setting $S = \mathbf{1} \in \mathbb{R}^D$ in Eq. (64) and then taking the derivative with respect to $\lambda$ gives:

$$\mathcal{L}'(\lambda) = \sum_{n=1}^{N} \frac{2}{(1 + \lambda - \|u_n\|^2)^2} \left( \frac{1}{1 + \lambda} u_n^T U^T Y - y_n \right) \tag{11}$$

$$* \left[ -\frac{(1 + \lambda)\|u_n\|^2}{1 + \lambda - \|u_n\|^2} \left( \frac{1}{1 + \lambda} u_n^T U^T Y - y_n \right) - u_n^T U^T Y \right]$$

Let $\hat{\varepsilon}_n$ be the scalars such that $y_n = \langle x_n, \hat{\theta} \rangle + \hat{\varepsilon}_n$. Letting $\hat{E} \in \mathbb{R}^N$ be the vector with entries $\hat{\varepsilon}_n$, we have $U^T \hat{E} = 0$, and so $U^T Y = U^T U \operatorname{diag}(\mathbf{1}) \hat{\theta} = \hat{\theta}$. Plugging this into Eq. (11) we get:

$$\mathcal{L}'(\lambda) = \sum_{n=1}^{N} \frac{2 \left( \frac{1}{1+\lambda} \langle u_n, \hat{\theta} \rangle - \langle u_n, \hat{\theta} \rangle - \hat{\varepsilon}_n \right)}{(\lambda + 1 - \|u_n\|^2)^2} \left[ -\frac{\|u_n\|^2 (1 + \lambda)}{\lambda + 1 - \|u_n\|^2} \left( \frac{1}{1 + \lambda} \langle u_n, \hat{\theta} \rangle - \langle u_n, \hat{\theta} \rangle - \hat{\varepsilon}_n \right) - \langle u_n, \hat{\theta} \rangle - \hat{\varepsilon}_n \right]$$

$$= \sum_{n=1}^{N} \frac{2 \left( -\frac{\lambda}{1+\lambda} \langle u_n, \hat{\theta} \rangle - \hat{\varepsilon}_n \right)}{(\lambda + 1 - \|u_n\|^2)^3} \left[ -\|u_n\|^2 (1 + \lambda) \left( -\frac{\lambda}{1 + \lambda} \langle u_n, \hat{\theta} \rangle - \hat{\varepsilon}_n \right) + (-\langle u_n, \hat{\theta} \rangle - \hat{\varepsilon}_n)(\lambda + 1 - \|u_n\|^2) \right]$$

Finally, rearranging to group terms multiplying $\lambda^2$ and $\lambda$, we get:

$$\mathcal{L}'(\lambda) = \frac{2}{(1 + \lambda)^4} \sum_{n=1}^{N} \frac{(1 + \lambda)^3}{(1 + \lambda - \|u_n\|^2)^3} \left( (1 - \|u_n\|^2) \langle u_n, \hat{\theta} \rangle^2 - \|u_n\|^2 \hat{\varepsilon}_n^2 + (1 - 2\|u_n\|^2) \hat{\varepsilon}_n \langle u_n, \hat{\theta} \rangle \right) \lambda^2$$

$$+ \left( (1 - \|u_n\|^2) \langle u_n, \hat{\theta} \rangle^2 - 2\|u_n\|^2 \hat{\varepsilon}_n^2 + (2 - 3\|u_n\|^2) \hat{\varepsilon}_n \langle u_n, \hat{\theta} \rangle \right) \lambda$$

$$+ \left( -\|u_n\|^2 \hat{\varepsilon}_n^2 + (1 - \|u_n\|^2) \hat{\varepsilon}_n \langle u_n, \hat{\theta} \rangle \right) \tag{12}$$

We can write this more compactly

$$\mathcal{L}'(\lambda) = \frac{2}{(1 + \lambda)^4} \sum_{n=1}^{N} \frac{(1 + \lambda)^3}{(1 + \lambda - \|u_n\|^2)^3} \left( \xi_{n1} \lambda^2 + \xi_{n2} \lambda + \xi_{n3} \right), \tag{13}$$

where the $\xi_i$ are defined by matching up coefficients between Eq. (12) and Eq. (13). Now we can prove the two main Lemmas needed to prove Theorem 1.

**Lemma 1.** *Take Assumptions 1 to 4. For a flat spectrum $S = \mathbf{1} \in \mathbb{R}^D$, there is some $\lambda_Q$ that is $O(1)$ such that $\mathcal{L}'(\lambda) = 0$ implies that $\lambda \in [0, \lambda_Q + o(1)]$.*

*Proof.* First, we can discard the $1/(1 + \lambda)^4$ in front of Eq. (13) for the purposes of deciding where $\mathcal{L}' = 0$; let $g(\lambda) = \mathcal{L}'(\lambda)(1 + \lambda)^4$:

$$g(\lambda) = \sum_{n=1}^{N} \frac{(1 + \lambda)^3}{(1 + \lambda - \|u_n\|^2)^3} \left( \xi_{n1} \lambda^2 + \xi_{n2} \lambda + \xi_{n3} \right), \tag{14}$$

Notice that $g$ is nearly a quadratic; in particular, if $\|u_n\|^2 = 0$, then $g$ is a quadratic. The idea is to let $\lambda_Q$ be the rightmost root of this quadratic; we then show that the perturbations away from this quadratic are small enough to imply that all zeros of $g$ lie in $[0, \lambda_Q + o(1)]$.

Write $\xi_{\cdot i} := \sum_n \xi_{ni}$. Then, via the quadratic formula, the roots of $g_Q(\lambda) := \xi_{\cdot 1} \lambda^2 + \xi_{\cdot 2} \lambda + \xi_{\cdot 3}$ are

$$\lambda = \frac{-\xi_{\cdot 2} \pm \left[ \xi_{\cdot 2}^2 - 4\xi_{\cdot 1} \xi_{\cdot 3} \right]^{1/2}}{2\xi_{\cdot 1}}. \tag{15}$$

We can apply the following facts from Lemma 4: $\xi_{\cdot 1}$ and $\xi_{\cdot 2}$ are $\Theta(1)$ and $\xi_{\cdot 3}$ is negative or $o(1)$. We can conclude that the positive root of Eq. (15) is larger than the negative root and is $O(1)$; call the positive root $\lambda_Q$. Now we need to bound the devitions of $g$ away from the quadratic $g_Q$. Let $\delta(\lambda) := g(\lambda) - g_Q(\lambda)$ be these deviations:

$$\delta(\lambda) := \sum_{n=1}^{N} \left( \left( \frac{\lambda + 1}{\lambda + 1 - \|u_n\|^2} \right)^3 - 1 \right) (\xi_{n1}\lambda^2 + \xi_{n2}\lambda + \xi_{n3}). \tag{16}$$

Notice that our quadratic $g_Q$ is convex, as $\xi_{\cdot 1} > 0$ by Lemma 4. Thus the way to move the roots of $g$ further right than $\lambda_Q$ is to have $\delta(\lambda)$ be negative. We can lower bound $\delta(\lambda) \geq \delta(0)$ by noting that $\xi_{\cdot 1}, \xi_{\cdot 2} > 0$. Thus:

$$\delta(\lambda) \geq \delta(0) = \sum_{n=1}^{N} \left( \frac{1}{(1 - \|u_n\|^2)^3} - 1 \right) \left( -\|u_n\|^2 \hat{\varepsilon}_n^2 + (1 - \|u_n\|^2)\hat{\varepsilon}_n \langle u_n, \hat{\theta} \rangle \right).$$

By Lemma 5, we have that $\delta(0) = o(1)$.

As we know $g(\lambda) = g_Q(\lambda) + \delta(\lambda) \geq g_Q(\lambda) + \delta(0)$, the final step of our proof is to find the right-most $\lambda$ for which $g_Q(\lambda) = -\delta(0)$, as beyond such a $\lambda$, $g > 0$. In fact, an upper bound on this $\lambda$ will suffice. Using convexity with the fact that $g_Q(\lambda_Q) = 0$, we have that beyond $\lambda = \lambda_Q + \delta(0)/g_Q'(\lambda_Q)$, $g_Q(\lambda) \geq \delta(0)$, and thus $g(\lambda) \geq 0$. If we knew that $g_Q'(\lambda_Q) = 2\lambda_Q\xi_{\cdot 1} + \xi_{\cdot 2}$ were $\Theta(1)$, we would be done, as:

$$\lambda_Q = \frac{\delta(0)}{g_Q'(\lambda_Q)} = \lambda_Q + \frac{\delta(0)}{2\lambda_Q\xi_{\cdot 1} + \xi_{\cdot 2}} = \lambda_Q + \frac{o(1)}{\Theta(1)} = \lambda_Q + o(1), \tag{17}$$

To see that $2\lambda_Q\xi_{\cdot 1} + \xi_{\cdot 2}$ is $\Theta(1)$, recall that we have $\lambda_Q$ is $O(1)$ and positive. And by Lemma 4, $\xi_{\cdot 1}$ and $\xi_{\cdot 2}$ are positive and $\Theta(1)$. We conclude that $2\lambda_Q\xi_{\cdot 1} + \xi_{\cdot 2} = \Theta(1)$. $\square$

**Lemma 2.** *Take Assumptions 1 to 4. Let $\lambda_Q$ be as defined in Lemma 1, and assume the covariate matrix $X$ has a flat spectrum $S = \mathbf{1} \in \mathbb{R}^D$. Then for any $\lambda \in [0, \lambda_Q + \Theta(1)]$ such that $\mathcal{L}'(\lambda) = 0$, it holds that $\mathcal{L}''(\lambda) > 0$.*

*Proof.* The strategy is similar to the proof of Lemma 1: we show that $\mathcal{L}''$ is nearly a quadratic, find the root of this quadratic, and then show that the location of this root can only change by $o(1)$ due to the deviations away from quadratic.

First, we need to compute $\mathcal{L}''$. Differentiating $\mathcal{L}'$ as given by Eq. (13) gives:

$$\mathcal{L}''(\lambda) = \frac{2}{1+\lambda} \sum_{n=1}^{N} \left( -\frac{\xi_{n1}\lambda^2 + \xi_{n2}\lambda + \xi_{n3}}{(1+\lambda)(1+\lambda - \|u_n\|^2)^3} - \frac{3(\xi_{n1}\lambda^2 + \xi_{n2}\lambda + \xi_{n3})}{(\lambda + 1 - \|u_n\|^2)^4} \right. \tag{18}$$

$$\left. + \frac{2\xi_{n1}\lambda + \xi_{n2}}{(\lambda + 1 - \|u_n\|^2)^3} \right) \tag{19}$$

Now, the first term in this sum is exactly $\mathcal{L}'(\lambda)$. By the conditions of the lemma, we have that this term sums to zero. Using this fact and some algebra, we can see that $\mathcal{L}''$ is also almost a quadratic:

$$\mathcal{L}''(\lambda) = \frac{2}{(1+\lambda)^5} \sum_{n=1}^{N} \frac{(1+\lambda)^4}{(1+\lambda - \|u_n\|^2)^4} \left( -\xi_{n1}\lambda^2 \right. \tag{20}$$

$$+ \left( 2(1 - \|u_n\|^2)\xi_{n1} - 2\xi_{n2} \right)\lambda \tag{21}$$

$$\left. + \left( (1 - \|u_n\|^2)\xi_{n2} - 3\xi_{n3} \right) \right), \tag{22}$$

where the $\xi_{ni}$'s are as defined in the proof of Lemma 1. As we are interested in the region where $\mathcal{L}'' > 0$, we can neglect the $1/(1+\lambda)^5$ factor in front; define $h(\lambda) := (1+\lambda)^5 \mathcal{L}''(\lambda)$. Now, define

the following

$$a_n := -\xi_{n1} \tag{23}$$
$$= 2\|u_n\|^2 \hat\varepsilon_n^2 - (1 - \|u_n\|^2)\langle u_n, \hat\theta\rangle^2$$

$$b_n := 2(1 - \|u_n\|^2)\xi_{n1} - 2\xi_{n2} \tag{24}$$
$$= \big(2(1 - \|u_n\|^2)^2 - (1 - \|u_n\|^2)\big)\langle u_n, \hat\theta\rangle^2 + \big(-2(1 - \|u_n\|^2)\|u_n\|^2 + 4\|u_n\|^2\big)\hat\varepsilon_n^2$$
$$+ \big(2(1 - \|u_n\|^2)(1 - 2\|u_n\|^2) - 2(2 - 3\|u_n\|^2)\big)\hat\varepsilon_n\langle u_n, \hat\theta\rangle$$

$$c_n := (1 - \|u_n\|^2)\xi_{n2} - 3\xi_{n3} \tag{25}$$
$$= (1 - \|u_n\|^2)^2\langle u_n, \hat\theta\rangle^2 + \big(-2(1 - \|u_n\|^2)\|u_n\|^2 + 3\|u_n\|^2\big)\hat\varepsilon_n^2$$
$$+ \big((1 - \|u_n\|^2)(2 - 3\|u_n\|^2) - 3(1 - \|u_n\|^2)\big)\hat\varepsilon_n\langle u_n, \hat\theta\rangle$$

$$h_Q(\lambda) := \sum_{n=1}^N a_n\lambda^2 + b_n\lambda + c_n \tag{26}$$

$$\delta^{(2)}(\lambda) := \sum_{n=1}^N \left(\left(\frac{1 + \lambda}{1 + \lambda - \|u_n\|^2}\right)^4 - 1\right)\left(a_n\lambda^2 + b_n\lambda + c_n\right). \tag{27}$$

Note that $h = h_Q + \delta^{(2)}$. Let $a_\cdot = \sum_n a_n$, and likewise for $b_\cdot, c_\cdot$. Application of Proposition 3 and Assumptions 1 and 3 gives:

$$c_\cdot = \xi_{\cdot 2} - 3\xi_{\cdot 3} + o(1). \tag{28}$$

In particular, noting that $\xi_{\cdot 3} < 0$ or is $o(1)$ and $\xi_{\cdot 2}$ is $\Theta(1)$ and positive, (Lemma 4), we have $c_\cdot > 0$ with $|c_\cdot| > |\xi_{\cdot 3}|$ for large enough $N$. Now, in general $h_Q$ will have two roots:

$$\lambda = \frac{-b_\cdot \pm [b_\cdot^2 - 4a_\cdot c_\cdot]^{1/2}}{2a_\cdot} = \frac{b_\cdot \mp [b_\cdot^2 + 4\xi_{\cdot 1}c_\cdot]^{1/2}}{2\xi_{\cdot 1}}.$$

Now, as $[b_\cdot^2 + 4\xi_{\cdot 1}c_\cdot]^{1/2} > b_\cdot$, only one of these roots is positive; call this root $\lambda'_Q$. We now want to show that $\lambda'_Q - \lambda_Q = \Theta(1)$ and is positive. We have that:

$$\lambda'_Q - \lambda_Q = \frac{b_\cdot + \xi_{\cdot 2} + [b_\cdot^2 + 4\xi_{\cdot 1}((1 - \|u_n\|^2)\xi_{\cdot 1} - 3\xi_{\cdot 3})]^{1/2} - [\xi_{\cdot 2}^2 - 4\xi_{\cdot 1}\xi_{\cdot 3}]^{1/2}}{\xi_{\cdot 1}} \tag{29}$$

We know the denominator is positive and $\Theta(1)$ by Lemma 4 so we just need to show the numerator is $\Theta(1)$ and positive. Combining Assumption 3 with the fact that $b_\cdot$ is positive or $o(1)$ (by Lemma 4), we have that the numerator satisfies:

$$\geq \xi_{\cdot 2} + [4\xi_{\cdot 1}^2 - 4\xi_{\cdot 1}^2 O(N^{-p}) - 12\xi_{\cdot 3}\xi_{\cdot 1}]^{1/2} - [\xi_{\cdot 2}^2 - 4\xi_{\cdot 1}\xi_{\cdot 3}]^{1/2}.$$

Now, as $\xi_{\cdot 1} = \Theta(1)$:

$$\geq \xi_{\cdot 2} + [\Theta(1) - 12\xi_{\cdot 3}\xi_{\cdot 1}]^{1/2} - \xi_{\cdot 2} - [|4\xi_{\cdot 1}\xi_{\cdot 3}|]^{1/2}.$$

By Lemma 4, $\xi_{\cdot 3}$ is either negative or $o(1)$. So the numerator satisfies:

$$\geq [\Theta(1) - 12\xi_{\cdot 3}\xi_{\cdot 1}]^{1/2} - [|4\xi_{\cdot 1}\xi_{\cdot 3}|]^{1/2},$$

which is $\Theta(1)$ and positive. Thus $\lambda'_Q - \lambda_Q$ is $\Theta(1)$ and is positive.

Finally, we need to lower bound $\delta^{(2)}(\lambda)$ on $[0, \lambda_Q + \Theta(1)]$. As $\lambda_Q = \Theta(1)$, Lemma 6 shows that $\delta^{(2)}(\lambda) = o(1)$ for all $\lambda \in [0, \lambda_Q + \Theta(1)]$. Thus for all $\lambda \in [0, \lambda_Q + O(1) - o(1)]$ for which $\mathcal{L}'(\lambda) = 0$, we have $h(\lambda) > 0$, and so $\mathcal{L}''(\lambda) > 0$. $\qquad\square$

### E.1 Technical Lemmas

**Lemma 3.** *Take real numbers $s_1, \ldots, s_N$ and $r_1, \ldots, r_N$, where $r_n \in [\ell, u]$ and $\sum_{n=1}^N s_n = 0$. Then:*

$$\left|\sum_{n=1}^N r_n s_n\right| \leq \frac{u - \ell}{2} \sum_{n=1}^N |s_n|.$$

*Proof.* Let $s_n^+ := \max(0, s_n)$ and $s_n^- := \max(0, -s_n)$. Then the condition $\sum_n s_n = 0$ implies that $\sum_n s_n^+ = \sum_n s_n^- = (1/2) \sum_n |s_n|$. So:

$$\left| \sum_{n=1}^{N} r_n s_n \right| \leq u \sum_{n=1}^{N} s_n^+ - \ell \sum_{n=1}^{N} s_n^- = \frac{u - \ell}{2} \sum_{n=1}^{N} |s_n|.$$

$\square$

Now we state a useful consequence of our above assumptions:

**Proposition 3.** *Take Assumptions 1 to 3. We have:*

$$\sum_{n=1}^{N} (1 - \|u_n\|^2) \hat{\varepsilon}_n \langle u_n, \hat{\theta} \rangle = o(1)$$

*Proof.* Notice that $\sum_n \hat{\varepsilon}_n \langle u_n, \hat{\theta} \rangle = \hat{E}^T U \hat{\theta} = 0$. So, we are trying to bound $\left| \sum_n \|u_n\|^2 \hat{\varepsilon}_n \langle u_n, \hat{\theta} \rangle \right|$. Assumption 3 implies that $\|u_n\|^2$ is $O(N^{-p})$. As $\|u_n\|^2$ is lower bounded by zero, we can apply Lemma 3 to get the upper bound:

$$\left| \sum_{n=1}^{N} \|u_n\|^2 \hat{\varepsilon}_n \langle u_n, \hat{\theta} \rangle \right| \leq O(N^{-p}) \sum_{n=1}^{N} |\hat{\varepsilon}_n \langle u_n, \hat{\theta} \rangle|. \tag{30}$$

By Cauchy-Schwarz, we can upper bound the sum as:

$$\sum_n |\hat{\varepsilon}_n \langle u_n, \hat{\theta} \rangle| \leq \left( \sum_{n=1}^{N} \hat{\varepsilon}_n^2 \right)^{1/2} \left( \sum_{n=1}^{N} \langle u_n, \hat{\theta} \rangle^2 \right)^{1/2} \tag{31}$$

By Assumption 1 and the fact that $\sum_n \langle u_n, \hat{\theta} \rangle^2 = \|\hat{\theta}\|^2$, we have overall:

$$\left| \sum_{n=1}^{N} \|u_n\|^2 \hat{\varepsilon}_n \langle u_n, \hat{\theta} \rangle \right| \leq O(N^{-p}) O(\sqrt{N}) \left\| \hat{\theta} \right\|^2 = o(1), \tag{32}$$

where the final equality holds because by assumption, $p > 1/2$ and $\|\hat{\theta}\| = O(1)$ $\square$

Our next lemma concerns the quadratic coefficients that show up in $\mathcal{L}'$ and $\mathcal{L}''$.

**Lemma 4.** *Recall the definitions of the coefficients of the quadratic parts of $\mathcal{L}'$ and $\mathcal{L}''$ from our proofs above:*

$$\xi_{n1} := \left(1 - \|u_n\|^2\right) \langle u_n, \hat{\theta} \rangle^2 - \|u_n\|^2 \hat{\varepsilon}_n^2 + \left(1 - 2\|u_n\|^2\right) \hat{\varepsilon}_n \langle u_n, \hat{\theta} \rangle$$

$$\xi_{n2} := \left(1 - \|u_n\|^2\right) \langle u_n, \hat{\theta} \rangle^2 - 2\|u_n\|^2 \hat{\varepsilon}_n^2 + \left(2 - 3\|u_n\|^2\right) \hat{\varepsilon}_n \langle u_n, \hat{\theta} \rangle$$

$$\xi_{n3} := -\|u_n\|^2 \hat{\varepsilon}_n^2 + \left(1 - \|u_n\|^2\right) \hat{\varepsilon}_n \langle u_n, \hat{\theta} \rangle$$

$$a_n := -\xi_{n1}$$
$$= 2\|u_n\|^2 \hat{\varepsilon}_n^2 - (1 - \|u_n\|^2) \langle u_n, \hat{\theta} \rangle^2$$

$$b_n := 2(1 - \|u_n\|^2) \xi_{n1} - 2\xi_{n2}$$
$$= \left(2(1 - \|u_n\|^2)^2 - (1 - \|u_n\|^2)\right) \langle u_n, \hat{\theta} \rangle^2 + \left(-2(1 - \|u_n\|^2)\|u_n\|^2 + 4\|u_n\|^2\right) \hat{\varepsilon}_n^2$$
$$+ \left(2(1 - \|u_n\|^2)(1 - 2\|u_n\|^2) - 2(2 - 3\|u_n\|^2)\right) \hat{\varepsilon}_n \langle u_n, \hat{\theta} \rangle$$

$$c_n := (1 - \|u_n\|^2) \xi_{n2} - 3\xi_{n3}$$
$$= (1 - \|u_n\|^2)^2 \langle u_n, \hat{\theta} \rangle^2 + \left(-2(1 - \|u_n\|^2)\|u_n\|^2 + 3\|u_n\|^2\right) \hat{\varepsilon}_n^2$$
$$+ \left((1 - \|u_n\|^2)(2 - 3\|u_n\|^2) - 3(1 - \|u_n\|^2)\right) \hat{\varepsilon}_n \langle u_n, \hat{\theta} \rangle$$

*Further, recall $\xi_{\cdot 1} := \sum_{n=1}^{N} \xi_{n1}$, and likewise for $\xi_{\cdot 2}, \xi_{\cdot 3}, a_\cdot, b_\cdot,$ and $c_\cdot$. The following statements hold:*

1. *$\xi_{\cdot 1}$ is positive and $\Theta(1)$.*

2. *$\xi_{\cdot 2}$ is positive and $\Theta(1)$.*

3. *Either $b_\cdot > 0$ or $b_\cdot$ is positive and $o(1)$.*

4. *Either $\xi_{\cdot 3} < 0$ or $\xi_{\cdot 3}$ is positive and $o(1)$.*

*Proof.* We prove each item below.

1. $\xi_{\cdot 1}$ is positive and $\Theta(1)$. Using Proposition 3, we have

$$\xi_{\cdot 1} = \|\hat{\theta}\|^2 - \sum_{n=1}^{N} \|u_n\|^2 (\langle u_n, \hat{\theta}\rangle^2 + \hat{\varepsilon}_n^2) + o(1) \tag{33}$$

$$\geq \|\hat{\theta}\|^2 - \sum_{n=1}^{N} \|u_n\|^2 (\langle u_n, \hat{\theta}\rangle^2 + 2\hat{\varepsilon}_n^2) + o(1) \tag{34}$$

$$= \Theta(1), \tag{35}$$

where in the final equality we have used Assumption 4.

2. $\xi_{\cdot 2}$ is positive and $\Theta(1)$. The proof of this is identical to the proof that $\xi_{\cdot 1}$ is positive and $\Theta(1)$ but with different constants.

3. Either $b_\cdot > 0$ or $b_\cdot$ is positive and $o(1)$. From the definition of $b_n$:

$$b_\cdot = \sum_{n=1}^{N} \|u_n\|^2 \hat{\varepsilon}_n^2 + \|u_n\|^2 \hat{\varepsilon}_n \langle u_n, \hat{\theta}\rangle - (\|u_n\|^2 - (\|u_n\|^2)^2)\langle u_n, \hat{\theta}\rangle^2$$
$$+ (\|u_n\|^2)^2 \hat{\varepsilon}_n^2 - (\|u_n\|^2 - 2(\|u_n\|^2)^2)\hat{\varepsilon}_n \langle u_n, \hat{\theta}\rangle$$

By Assumption 3, any term with $(\|u_n\|^2)^2$ sums up to $o(1)$. Thus:

$$b_\cdot = o(1) + \sum_{n=1}^{N} \|u_n\|^2 \hat{\varepsilon}_n^2 - \|u_n\|^2 \langle u_n, \hat{\theta}\rangle^2 \tag{36}$$

$$= \left( \sum_{n=1}^{N} \|u_n\|^2 \hat{\varepsilon}_n^2 \right) - O(N^{-p}) \tag{37}$$

Thus we have that $b_\cdot$ is either positive or is $o(1)$.

4. Either $\xi_{\cdot 3} < 0$ or $\xi_{\cdot 3}$ is positive and $o(1)$. By Proposition 3

$$\xi_{\cdot 3} = o(1) - \sum_{n=1}^{N} \|u_n\|^2 \hat{\varepsilon}_n^2. \tag{38}$$

So $\xi_{\cdot 3}$ is either $o(1)$ and positive or is negative.

$\square$

**Lemma 5.** *Take Assumptions 1 to 3. We have:*

$$\sum_{n=1}^{N} \left( \frac{1}{(1 - \|u_n\|^2)^3} - 1 \right) \left( -\|u_n\|^2 \hat{\varepsilon}_n^2 + (1 - \|u_n\|^2)\hat{\varepsilon}_n \langle u_n, \hat{\theta}\rangle \right) = o(1). \tag{39}$$

*Proof.* We first show that

$$-\sum_{n=1}^{N} \left( \frac{1}{(1 - \|u_n\|^2)^3} - 1 \right) \|u_n\|^2 \hat{\varepsilon}_n^2 = o(1).$$

First, note that as $0 \leq \|u_n\|^2 < 1$, this quantity is strictly negative. So, it suffices to lower bound it by a quantity that is $o(1)$. We apply the lower bound

$$-\sum_{n=1}^{N} \left( \frac{1}{(1 - \|u_n\|^2)^3} - 1 \right) \|u_n\|^2 \hat{\varepsilon}_n^2 \geq -\left( \frac{1}{(1 - \|u_{\max}\|^2)^3} - 1 \right) \|u_{\max}\|^2 \sum_{n=1}^{N} \hat{\varepsilon}_n^2.$$

By a Taylor expansion around $\|u_{\max}\|^2 = 0$, we have:

$$= -\left( \|u_{\max}\|^2 + 6(\|u_{\max}\|^2)^2 + O((\|u_{\max}\|^2)^3) - \|u_{\max}\|^2 \right) \sum_{n=1}^{N} \hat{\varepsilon}_n^2.$$

By Assumptions 1 and 3, this is equal to $O(N^{-2p})O(N) = o(1)$.

Next we show that

$$\sum_{n=1}^{N} \left( \frac{1}{(1 - \|u_n\|^2)^3} - 1 \right) \left( 1 - \|u_n\|^2 \right) \hat{\varepsilon}_n \langle u_n, \hat{\theta} \rangle = o(1).$$

To start, note that $\sum_n \hat{\varepsilon}_n \langle u_n, \hat{\theta} \rangle = \hat{E}^T U^T \hat{\theta} = 0$. We can then apply Lemma 3 to upper bound the absolute value of our quantity of interest:

$$\left| \sum_{n=1}^{N} \left( \frac{1}{(1 - \|u_n\|^2)^3} - 1 \right) \left( 1 - \|u_n\|^2 \right) \hat{\varepsilon}_n \langle u_n, \hat{\theta} \rangle \right|$$

$$\leq \left( \frac{1}{(1 - \|u_{\max}\|^2)^3} - 1 \right) \left( 1 - \|u_{\max}\|^2 \right) \sum_{n=1}^{N} |\hat{\varepsilon}_n \langle u_n, \hat{\theta} \rangle| \tag{40}$$

Now, by a Taylor expansion of the quantity outside the sum around $\|u_{\max}\|^2 = 0$

$$= \left( 0 + 3\|u_{\max}\|^2 + O((\|u_{\max}\|^2)^2) \right) \sum_{n=1}^{N} |\hat{\varepsilon}_n \langle u_n, \hat{\theta} \rangle|. \tag{41}$$

Applying Cauchy Schwarz along with Assumption 3:

$$\leq O(N^{-p}) \left( \sum_{n=1}^{N} \hat{\varepsilon}_n^2 \right)^{1/2} \left( \sum_{n=1}^{N} \langle u_n, \hat{\theta} \rangle^2 \right)^{1/2} \tag{42}$$

Applying Assumption 1 and then Assumption 2:

$$= O(N^{-p})O(N^{1/2})\|\hat{\theta}\|^2 = o(1). \tag{43}$$

$\square$

**Lemma 6.** *Take Assumptions 1 to 3 and as in the proof of Lemma 2, define:*

$$\delta^{(2)}(\lambda) := \sum_{n=1}^{N} \left( \left( \frac{1 + \lambda}{1 + \lambda - \|u_n\|^2} \right)^4 - 1 \right) (a_n \lambda^2 + b_n \lambda + c_n), \tag{44}$$

*where $a_n, b_n, c_n$ are as defined in the proof of Lemma 2. Then, for $\lambda \in [0, c]$, where $c > 0$ is some constant in $N$, we have that $\delta^{(2)}(\lambda) = o(1)$.*

*Proof.* First, we have:

$$\left( \frac{1 + \lambda}{1 + \lambda - \|u_n\|^2} \right)^4 - 1 \leq \left( \frac{1 + \lambda}{1 + \lambda - \|u_{\max}\|^2} \right)^4 - 1 \tag{45}$$

$$= \left( 0 + 4\|u_{\max}\|^2 + O((\|u_{\max}\|^2)^2) \right) \tag{46}$$

$$= O(N^{-p}) \tag{47}$$

where the second equality holds by a Taylor expansion around $\|u_{\max}\|^2 = 0$, and the third equality holds by Assumption 3. Now, we can bound $\delta^{(2)}(\lambda)$ as:

$$\delta^{(2)}(\lambda) \leq O(N^{-p}) \sum_{n=1}^{N} |a_n \lambda^2 + b_n \lambda + c_n| \leq O(N^{-p}) \sum_{n=1}^{N} |a_n| + |b_n| + |c_n|, \tag{48}$$

where the second inequality is a result of $\lambda \leq c = O(1)$. We now bound the sums $\sum_n |a_n|$, $\sum_n |b_n|$, and $\sum_n |c_n|$ to complete the proof.

$$\sum_{n=1}^{N} |a_n| \leq \sum_{n=1}^{N} 2\|u_n\|^2 \hat{\varepsilon}_n^2 + (1 - \|u_n\|^2)\langle u_n, \hat{\theta} \rangle^2 \tag{49}$$

$$= O(N^{1-p}) + (1 - O(N^{-p})) \left\| \hat{\theta} \right\|^2 \tag{50}$$

$$= O(N^{1-p}), \tag{51}$$

where the first line holds by the definition of $a_n$, the next by Assumptions 1 and 3, and the third by Assumption 2. We continue to bound:

$$\sum_{n=1}^{N} |b_n| \tag{52}$$

$$\leq \sum_{n=1}^{N} 2(1 - \|u_n\|^2)^2 \langle u_n, \hat{\theta} \rangle^2 + 4\|u_n\|^2 \hat{\varepsilon}_n^2 + (2 + 4(\|u_n\|^2)^2 + 6\|u_n\|^2)|\hat{\varepsilon}_n \langle u_n, \hat{\theta} \rangle| \tag{53}$$

$$= (2 + O(N^{-p}) \left\| \hat{\theta} \right\|^2 + O(N^{1-p}) + (2 + O(N^{-p})) \left( \sum_{n=1}^{N} \hat{\varepsilon}_n^2 \right)^{1/2} \left( \sum_{n=1}^{N} \langle u_n, \hat{\theta} \rangle^2 \right)^{1/2} \tag{54}$$

$$= O(\sqrt{N}), \tag{55}$$

where the first line is by definition of $b_n$, the second is by Assumptions 1 and 3 and the Cauchy-Schwarz inequality, and the third is by Assumptions 1 and 2. We continue by bounding:

$$\sum_{n=1}^{N} |c_n| \tag{56}$$

$$\leq \sum_{n=1}^{N} (1 - \|u_n\|^2)^2 \langle u_n, \hat{\theta} \rangle^2 + 4\|u_n\|^2 \hat{\varepsilon}_n^2 + (2 + 3(\|u_n\|^2)^2 + 3\|u_n\|^2)|\hat{\varepsilon}_n \langle u_n, \hat{\theta} \rangle| \tag{57}$$

$$\leq \left\| \hat{\theta} \right\|^2 + O(N^{1-p}) + (2 + O(N^{-p}) \left( \sum_{n=1}^{N} \hat{\varepsilon}_n^2 \right)^{1/2} \left( \sum_{n=1}^{N} \langle u_n, \hat{\theta} \rangle^2 \right)^{1/2} \tag{58}$$

$$= O(\sqrt{N}), \tag{59}$$

where the line-by-line reasoning is the same as that of our bound on $\sum_n |b_n|$ above. Plugging our bounds into Eq. (48), we get that for all $\lambda \in [0, c]$, we have that $\delta^{(2)}(\lambda) \leq O(N^{-p})O(\sqrt{N}) = o(1)$, as $p > 1/2$ by Assumption 3. $\qquad \square$

## F  Proof of Corollary 1

We first state a theorem about the concentration of i.i.d. sub-Gaussian matrices. Let $s_D \leq \cdots \leq s_1$ be the singular values of $X$.

**Theorem 2** (Theorem 4.6.1 from Vershynin [2018]). *Suppose that the $x_n$ are independent sub-Gaussian isotropic random vectors with maximum sub-Gaussian constant $K$. Then for some constant $C > 0$ and any $t \geq 0$, the following holds with probability at least $1 - 2e^{-t^2}$:*

$$\sqrt{N} - CK^2(\sqrt{D} + t) \leq s_D \leq s_1 \leq \sqrt{N} + CK^2(\sqrt{D} + t). \tag{60}$$

We now restate and then prove Corollary 1.

**Corollary 1.** *Take Assumptions 3 and 4. Assume we have a well-specified linear model for some $\theta^* \in \mathbb{R}^D$; that is, $y_n = \langle x_n, \theta^* \rangle + \varepsilon_n$, where $\varepsilon_n \overset{i.i.d.}{\sim} \mathcal{N}(0, \sigma^2)$. If $\sigma$ is sufficiently small, $\hat{\theta}$ is consistent for $\theta^*$, and the entries of the covariate matrix $x_{nd}$ are i.i.d. sub-Gaussian random variables, then $\mathcal{L}$ is quasiconvex with probability tending to 1 as $N \to \infty$.*

*Proof.* The idea is to show that our assumptions imply that Assumptions 1 and 2 hold, as well as that the spectrum of $X$ becomes uniform with high probability. Our assumption that $\hat{\theta}$ is consistent for $\theta^*$ immediately implies that $\|\hat{\theta}\| = O(1)$. Next, as discussed after Assumption 1, we can stack the $\varepsilon_n$ into a vector $E \in \mathbb{R}^N$. We then have that $\|E\|^2 \geq \|(I_N - UU^T)E\|^2 = \|\hat{E}\|^2 = \sum_n \hat{\varepsilon}_n^2$. As $\|E\|^2 = O(N)$ with probability tending towards 1 as $N \to \infty$, we have that Assumption 1 holds with probability tending towards 1 as $N \to \infty$.

Now, by Theorem 2 with $t = N^{1/3}$ (any $t$ that goes to infinity with $N$ but is $o(\sqrt{N})$ will work), we have that the singular values of $X$ satisfy

$$\sqrt{N} - o(\sqrt{N}) \leq s_D \leq s_1 \leq \sqrt{N} + o(\sqrt{N}). \tag{61}$$

with probability at least $1 - o(1)$. By Proposition 2, the quasiconvexity of $\mathcal{L}$ is invariant to a scaling of the singular values; thus we can divide all singular values by $\sqrt{N}$ to get that all singular values lie in the interval $[1 - o(1), 1 + o(1)]$ with probability going to 1. Thus, for any neighborhood $\Delta$ of $\mathbf{1} \in \mathbb{R}^D$, the (normalized) singular values will eventually lie within $\Delta$ with arbitrariliy high probability. Thus all of the conditions of Theorem 1 are met, implying that $\mathcal{L}$ will be quasiconvex with probability tending towards 1 as $N \to \infty$.

$\square$

## G Generating zero-mean orthonormal matrices uniformly at random

In our experiments in Section 6, we draw $N \times D$ orthonormal matrices $U$ with zero-mean columns from the uniform distribution over such matrices. To do so, we generate vectors $a_1, \ldots, a_D \in \mathbb{R}^D$ such that $a_{nd} \overset{i.i.d.}{\sim} \mathcal{N}(0, 1)$. We then use the Gram-Schmidt process to orthogonalize the vectors $\{\mathbf{1}, a_1, \ldots, a_D\}$; the second through $D + 1$th outputted vectors make up the columns of $U$. Notice this procedure requires $N < D$.

In our experiments surrounding Assumption 1, we need to generate vectors $R \in \mathbb{R}^N$ such that $U^T R = 0$ uniformly over such $R$'s. To do so, we generate a vector $a \sim \mathcal{N}(0, I_N)$. We then compute $b = (I_N - UU^T)a$; setting $R = b/\|b\|$ yields the result.

Why is this uniform over all vectors in the null space of $U$? Recall that $a$ is an isotropic random vector. It is well-known that this implies that $a/\|a\|$ is uniform over the unit sphere. As multiplication by $I_N - UU^T$ is an orthogonal projection, we have that $b = (I_N - UU^T)a$ is isotropic over the null space of $U$. Thus $b/\|b\|$ is uniform over the null-space of $U$. The same reasoning shows that the use of the Gram-Schmidt algorithm to generate orthonormal zero-column-mean matrices $U$ is uniform over such matrices – we start with isotropic random vectors, Gram Schmidt applies orthogonal projections to each and then normalizes the results.

## H Replicating experiments with error bars

For each of the experiments in Fig. 3 (our experiments about the size of the neighborhood $\Delta$ and $U$'s violating Assumption 3), we repeat the experiment five times to understand the random variability in each experiment. Fig. 8 shows the result. All plots are created exactly as in Fig. 3, except each dot is now an average over all five trials. Error bars are equal to two times the standard deviation across these five trials. We see that our conclusions from Fig. 3 still hold. In some cases, the error bars are so small that they are barely visible on the scale of these plots.

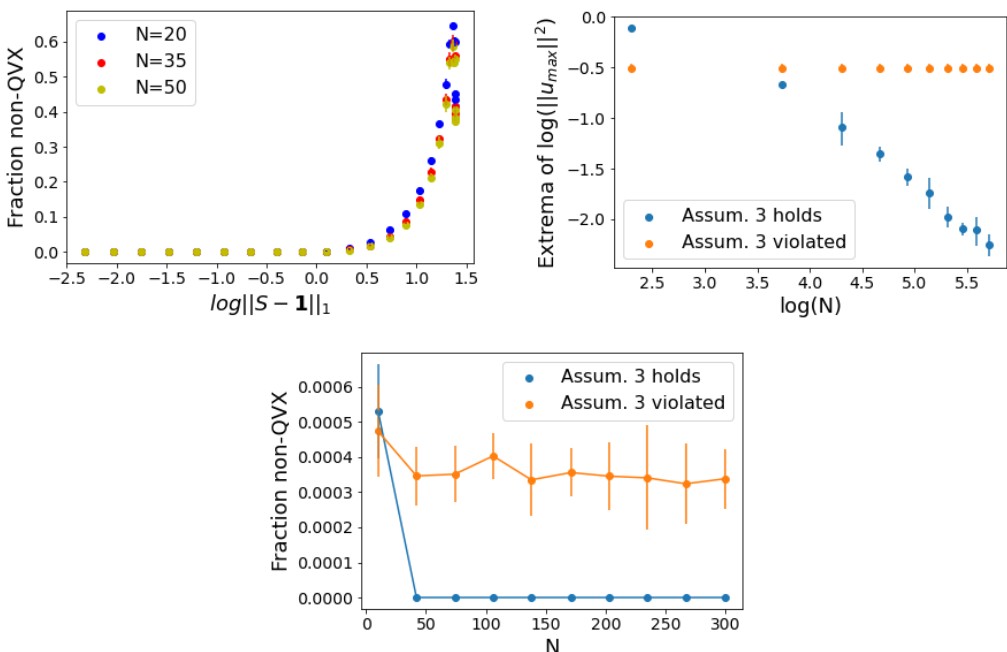

Figure 8: Replication of Fig. 3 with error bars; we repeat the original caption here for reading convenience. (*Upper left*): We generate many datasets and plot the fraction that are not quasiconvex, varying $N$ and the distance of the spectrum from uniformity ($\|S - \mathbf{1}\|_1$). (*Upper right*): We generate two sets (orange, blue) of left-singular vector matrices $U$. In the blue case, we check that the maximum of $\log \|u_{\max}\|^2$ across all $U$ for a particular $N$ decreases roughly linearly on a log-log plot (i.e. the blue set satisfies Assumption 3). In the orange case, we check that the minimum of $\log \|u_{\max}\|^2$ across all $U$ for a particular $N$ is roughly constant (i.e. the orange set does not satisfy Assumption 3). (*Lower*): For all the $U$ matrices from the upper right plot, we generate many datasets and plot the fraction that are not quasiconvex.

# I  Implementation details of experiments

## I.1  Efficiently computing $\mathcal{L}(\lambda)$

The LOOCV loss is given in Eq. (2) of the main text as:

$$\mathcal{L}(\lambda) := \sum_{n=1}^{N} \left( \langle x_n, \hat{\theta}^{\backslash n}(\lambda) \rangle - y_n \right)^2, \tag{62}$$

While this seems to require solving $N$ regression problems to obtain $\hat{\theta}^{\backslash n}(\lambda)$ for $n = 1, \dots, N$, there is a well-known explicit formula for $\mathcal{L}$ that makes use of the Sherman-Morrison formula:

$$\mathcal{L}(\lambda) = \sum_{n=1}^{N} \frac{1}{(1 - Q_n)^2} \left( \langle x_n, \hat{\theta} \rangle - y_n \right)^2, \tag{63}$$

where $Q_n := x_n^T \left( X^T X + \lambda I_D \right)^{-1} x_n$.

Using the results of Proposition 2, we know that the matrix of right singular vectors $V$ does not matter for the quasiconvexity of $\mathcal{L}$. So, in all of our experiments, we set $V = I_D$, which further simplifies Eq. (63). In particular, letting $u_n$ be the $n$th row of $U$, we have $x_n = u_n \operatorname{diag}(S)$ and $Q_n = u_n^T \operatorname{diag}(S^2/(S^2 + \lambda)) u_n$. Then:

$$\mathcal{L}(\lambda) = \sum_{n=1}^{N} \frac{1}{\left( 1 - u_n^T \operatorname{diag}\left( \frac{S^2}{S^2 + \lambda} \right) u_n \right)^2} \left( u_n^T \operatorname{diag}\left( \frac{S^2}{S^2 + \lambda} \right) U^T Y - y_n \right)^2. \tag{64}$$

We are given $U$ and $S$ in all of our experiments, so this formula is much faster to compute than Eq. (63), as it requires no matrix inversions.

## I.2   Numerically checking for quasiconvexity

To check for quasiconvexity in our experiments, we evaluate $\mathcal{L}$ on a dense, regularly spaced grid $\lambda_1, \ldots, \lambda_T$. In this appendix, we describe how, given $\mathcal{L}(\lambda_1), \ldots, \mathcal{L}(\lambda_T)$, we check whether $\mathcal{L}$ is quasiconvex. In short, we numerically check whether $\mathcal{L}'$ ever switches from positive to negative (the condition for a local maximum); note that this must occur inbetween any two local minima, so we do not have to count the number of local minima. To approximate the sign of $\mathcal{L}'$, we use $s_i := \mathrm{Sign}(\mathcal{L}(\lambda_{i+1} - \mathcal{L}(\lambda_i)))$ for $i = 1, \ldots, T-1$. We then report quasiconvexity if there exists any $i$ for which $s_i = 1$ and $s_{i+1} = -1$; that is, if our approximation to the derivative changes from positive to negative.

Note there are two ways in which this procedure can fail. The first is that our grid of $\lambda_t$'s may be insufficiently dense to capture non-quasiconvex behavior; however, in practice, we have never observed this to be an issue, as we have never found increasing the density of the grid of $\lambda$'s to reveal extra local minima. The second issue is that non-quasiconvex behavior may occur beyond the maximal $\lambda_T$ we specify. We again do not have an exact way of preventing this in practice. However, in all of our experiments, we specify a $\lambda_T$ that is orders of magnitude larger than the maximal singular value of $X$. At this point, we expect $\mathcal{L}(\lambda)$ to be an essentially flat function; thus, any missed local minima are likely of effectively the same value as $\mathcal{L}(\infty)$. In any case, if either failure mode were occurring in our of our experiments, fixing it would only make the (already concerning) non-quasiconvexity in our experiments look more severe.