# OpenReview forum: "Can we globally optimize cross-validation loss? Quasiconvexity in ridge regression"
_NeurIPS.cc/2021/Conference — NeurIPS 2021 Poster_

### Official Review · Reviewer_tmqa · 2021-07-10

**Rating:** 6
**Confidence:** 3

**Summary:**

The manuscript characterizes conditions under which the leave-one-out cross-validation error of ridge regression as a function of the ridge hyper-parameter has quasi-convex shape. The quasi-convexity would make the search for an optimal hyper-parameter value easier due to its single global optimum. Provided that certain mild technical assumptions about the data hold, quasi-convexity tends to emerge when the singular value spectrum of the training data is uniform enough. The effect of the assumptions and their violations are experimentally analysed with simulations.

**Limitations And Societal Impact:**

Limitations are discussed properly and there are no negative societal impacts.

**Main Review:**

The manuscript is clearly written which made reading enjoyable, and I did not spot any problems with the math or experimental analysis. To my knowledge, the presented detailed analysis of the quasi-convexity conditions of the ridge regression's leave-one-out cross-validation error as a function of the ridge hyper-parameter is original work and the observations are interesting at least from the theoretical point of view. Perhaps the biggest issue is that the results may not yet form a practical enough entity that would provide comprehensive guidelines for CV based ridge selection, such that would also cover cases when the considered conditions do not apply, especially the non-uniform spectra. Further studies and results, for example, about the number of severity of the possible local optima, are anticipated in the discussion section, and the current manuscript is indeed a good step towards that direction.

As a particular detail that would improve the paper, while the authors refer to several articles showing that CV is a good proxy for the out-of-sample error, I would still prefer to also see the out-of-sample error plots as a function of ridge parameter in addition to the CV plots, especially when there is a large enough data available for testing, because that would illuminate better how much practical importance it would have to find the global optimum of CV curve rather than a local one. Moreover, as said above, since the motivation for the study is very practical, I would prefer to see at least some sort of guidelines about how to carry out ridge selection with CV, given the results of this paper. For example, how to confirm that the conditions hold and then how to do the search when they hold and when they do not.




**Time Spent Reviewing:**

Can not estimate as the work was distributed into too many distinct time slots.

---

> ### Author Response · Authors · 2021-08-10
> **Reviewer Response**
>
> We are excited to hear that the reviewer found our paper enjoyable. In addition to our general comment to all reviewers, we respond to specific points below.
>
> **“I would still prefer to also see the out-of-sample error plots as a function of ridge parameter in addition to the CV plots”** Please see our discussion in the general comment.
>
> **“I would prefer to see at least some sort of guidelines about how to carry out ridge selection with CV, given the results of this paper. For example, [(1)] how to confirm that the conditions hold and then [(2)] how to do the search when they hold and when they do not.”**
> (1) We first consider how to check that our conditions hold. For completeness, we consider two interpretations of this sentence (1.A and 1.B below).
> (1.A) One interpretation asks: How should we computationally access the spectrum of X? There are many ways to do this, and we will include a discussion of some of these options in a revision:
> (1.A.i) When N and D are small (as in our examples), you can directly compute the SVD to verify the spectrum is near uniform.
> (1.A.ii) When D is small (and for any N), you can form X^(T)X and estimate its largest and smallest eigenvalue (and thereby compute the condition number, and bound the non-uniformity of the spectrum) efficiently via Rayleigh quotient iteration (see Trefthen and Bau 1997).
> (1.A.iii) If N>>D, you may wish to avoid forming the Gram matrix. In this case, you can use randomized sketching to get a (1+- epsilon) approximation to the singular values in O(min(N*D*log(D)+D^(3),nnz(X)+D^4)) time [Woodruff 2014].
> (1.A.iv) If N~D are both large, you may wish to use spectral density estimation [Lin et al. 2016], which gives an estimate for the density of a matrix’s eigenvalues, and has been shown to scale to large problems [Ghorbani et al. 2019, Yao et al. 2020].
>
> (1.B) Alternatively, the reviewer may mean to ask: given the spectrum of X, how can we know it is close enough to uniform for Theorem 1 to hold? Trying to answer this question is the purpose of our first experiment in the top-left of Figure 3. Here, we see that, for even modest $N$, as long as $\|S - 1\|_1 <= 10$, we can expect no non-quasiconvexity.
>
> (2) Next, we consider what to do when the conditions hold and when they do not. First, consider the case when our results indicate that the LOOCV objective may be non-quasiconvex, and thus may have local optima. In general, optimization of a multimodal objective is NP-hard, so our recommendation would be to use multiple random restarts (in the case of gradient descent) or a very large grid (in the case of grid search); practitioners will have to hope that these steps are sufficient. When our results indicate that the LOOCV objective is quasiconvex, we defer to pre-existing advice about how to efficiently optimize the LOOCV objective (e.g. Pedregosa 2016).
>
> **References:**
>
> Ghorbani, Behrooz, Shankar Krishnan, and Ying Xiao. "An investigation into neural net optimization via hessian eigenvalue density." In International Conference on Machine Learning, pp. 2232-2241. PMLR, 2019.
>
> Lin, Lin, Yousef Saad, and Chao Yang. "Approximating spectral densities of large matrices." SIAM review 58, no. 1 (2016): 34-65.
>
> Trefethen, Lloyd N., and David Bau III. Numerical linear algebra. Vol. 50. Siam, 1997.
>
> Woodruff, David P. "Sketching as a Tool for Numerical Linear Algebra." Foundations and Trends® in Theoretical Computer Science 10, no. 1–2 (2014): 1-157.
>
> Yao, Zhewei, Amir Gholami, Kurt Keutzer, and Michael W. Mahoney. "Pyhessian: Neural networks through the lens of the hessian." In 2020 IEEE International Conference on Big Data (Big Data), pp. 581-590. IEEE, 2020.

---

> > ### Comment · Reviewer_tmqa · 2021-08-31
> > **Reviewer response**
> >
> > I thank the authors for the detailed answer. Even better if the descriptions and guidelines will also get included in the final paper in some form.

---

> > > ### Author Response · Authors · 2021-09-01
> > > **Thank you for the reply**
> > >
> > > Thank you for reading through our response! We will definitely update the text of our paper.

---

### Official Review · Reviewer_YgJz · 2021-07-14

**Rating:** 7
**Confidence:** 3

**Summary:**

This paper demonstrates that leave-one-out cross-validation (LOOCV) risk estimate for ridge regression can have multiple local minima (both in real data and in simulations). It then goes on to prove that under some restrictive set of conditions the LOOCV loss is quasiconvex and only has one local=global minimum.

The paper is easy to follow and fun to read. The fact that LOOCV curve can have local minima was pretty surprising to me (but I may be unaware of some relevant literature). Not sure it's a perfect fit specifically for NeurIPS, but I vote for acceptance.

**Limitations And Societal Impact:**

Not discussed (but not really needed in this case, imho).

**Main Review:**

This paper demonstrates that leave-one-out cross-validation (LOOCV) risk estimate for ridge regression can have multiple local minima (both in real data and in simulations). It then goes on to prove that under some restrictive set of conditions the LOOCV loss is quasiconvex and only has one local=global minimum.

The paper is easy to follow and fun to read. The fact that LOOCV curve can have local minima was pretty surprising to me (but I may be unaware of some relevant literature). Not sure it's a perfect fit specifically for NeurIPS, but I vote for acceptance.


MAJOR ISSUES

* The paper claims that LOOCV loss may fail to be quasiconvex in the n<p regime (N<D in the paper notation). I would like to see the following questions at least discussed in the Discussion (perhaps for some of them the authors can even show non-quasiconvexity example in a simulation?):

  a) Can the true out-of-sample risk be non-quasiconvex?
  b) Can the k-fold CV loss be non-quasiconvex (for k \ne N)?
  c) Can the LOOCV loss be non-quasiconvex when n>p?

  (a) and (c) are most interesting. For (c), there is recent work showing that surprising phenomena can happen in that regime: https://jmlr.org/papers/v21/19-844.html. I wonder how this situation manifests itself with LOOCV. That would be worth discussing.

* It's a bit unclear if the authors think quasiconvexity is common or not *in practice*. Line 54 seems to suggest that it is rare (i.e. assumptions to prove quasiconvexity are common). However, in my experience, most often the spectrum of X is very non-flat and fast-decaying, with strong initial singular values. This goes against the assumptions needed to prove quasiconvexity, so I would actually expect non-quasiconvexity to be the typical situation in practice then, is that right? A bit surprising then that nobody noticed it until now...

* Section 2 fails to mention the standard explicit formula for LOOCV risk (1\n \sum e_i^2 / (1-h_i)^2). Did the authors actually use LOOCV for experiments (i.e. fit N separate models)? I guess not -- so this formula should be in here.

* Experiments shown in Figure 1b-c are interesting, and it would be interesting to have a fuller investigation described in the text: line 106 says "for many values of R" -- for which? What happens for R on a grid from 1 to 20? line 110 says "a subset of size N=50". Which subset? Random? How often does one get non-quasiconvexity for N=50? What about other N sizes? This does not have to be a figure, but at least some fuller empirical description in the text.

* Whenever the authors say that they checked quasiconvexity (as in Figure 2) -- how was it done? This is never described.


MINOR ISSUES

* line 25: this recent paper http://proceedings.mlr.press/v130/patil21a.html should probably be cited here too. And it may make sense to look into the references cited in there to see if there are other important citations.

* line 34: here and below "gradient-based" optimisation is mentioned a few times. I am not aware of anybody optimizing ridge penalty using gradient descent. Is there an expression for the gradient? If not, how would one implement it?

* Figure 1: fonts are too small. Make sure the figures are readable when printed out.

* line 37 -- "a" is a typo.

* Figure 2: maybe label axes in units of \pi?

* line 142-143: this sounds very naive. Proposition 2 is almost obvious, and it is clear that U and S are actually the variables of interest here.

* Figure 3: panel 2 goes together with panel 3. I would put them in one row, or even better split this figure into two separate ones.

* line 257: use \cdot instead of *

**Time Spent Reviewing:**

3

---

> ### Author Response · Authors · 2021-08-10
> **Reviewer response**
>
> We are excited to hear that the reviewer found our paper readable, fun, and informative -- and we thank the reviewer for their careful reading of the paper. In addition to our comment to all reviewers, we respond to specific points below.
>
> **“The paper claims that LOOCV loss may fail to be quasiconvex in the $n<p$ regime ($N<D$ in the paper notation)"** We briefly clarify here that our empirical and theoretical results are for the N > D regime; see the beginning of Section 2.
>
> **“a) Can the true out-of-sample risk be non-quasiconvex? b) Can the k-fold CV loss be non-quasiconvex”** We think these are very interesting questions! We discuss the point (a) in a general comment to all reviewers. In both cases: while a full theoretical examination may be out of scope of the current paper, we will plan to include illustrative experiments in a revision.
>
> **“It's a bit unclear if the authors think quasiconvexity is common or not in practice.”** We agree that our existing text is unclear on this point; thank you for pointing this out. Our conjecture is that non-quasiconvexity is not common in practice, but that the results of our paper imply it can occur. In particular, we suspect that as N gets larger, one can have a larger gap in the spectrum of X and still be very likely to obtain quasiconvexity. For example, if we make Figure 2 with N = 20, we find that the yellow regions shrink; likewise, when we reconstruct Figure 1 with larger N, we find less non-quasiconvexity. We will make this conjecture clear in a revision -- and discuss this supporting evidence.
>
> **“Section 2 fails to mention the standard explicit formula for LOOCV risk (1\n \sum e_i^2 / (1-h_i)^2).”** Thanks for this catch. Technically, we used a simplified version of this formula that accounts for the fact that V = the identity matrix to speed up our simulation experiments. We will be sure to include these details in a revision.
>
> **“Whenever the authors say that they checked quasiconvexity (as in Figure 2) -- how was it done? This is never described.”** To check for quasiconvexity, we evaluated L on a dense grid. We then examined sign changes in the pairwise differences L(lambda + delta) - L(lambda). This is not entirely non-trivial to implement in a numerically stable way; we will be sure to include this information in an appendix. Also we will be sure to add a discussion of the pros and cons of this procedure; note that detecting non-quasiconvexity is conclusive in this procedure but it is possible to miss some forms of non-quasiconvexity (a point that relates to our general grid search discussion).
>
> **Minor issues:**
> The reviewer suggests a citation and some additional discussion of gradient descent, and points to typos / stylistic improvements. We again thank the reviewer for their detailed feedback and will make these changes in a revision.

---

> > ### Comment · Reviewer_YgJz · 2021-08-13
> > **Thanks**
> >
> > Thanks for you reply.
> >
> > (Of course you are right, and when I wrote "n<p" in my first major issue above, I actually meant "n>p", and vice versa. The JMLR paper I mentioned is about the underdetermined case n<p; it shows that the optimal ridge penalty in this situation can be 0 or even negative.)

---

### Official Review · Reviewer_MpnJ · 2021-07-16

**Rating:** 4
**Confidence:** 5

**Summary:**

The authors consider the problem of analyzing the shape of the leave-one-out cross-validation loss, and establish sufficient conditions in terms of properties of the observation matrix $X$ under which the loss is quasi-convex, in a finite sample setting with assumptions related to the Fisherian ($n \rightarrow \infty$, $p$ fixed) asymptotic. The theoretical results are illustrated through some simulations.

**Limitations And Societal Impact:**

The main limitation of this work are centered around the $p$ fixed $n \rightarrow \infty$ setup, which is of limited interest in the context of penalized regression. In particular, this limitation is apparent in the assumption that the empirical spectral distribution of the Gram matrix $X^\top X$ must be close to a point mass at 1. I would hope that the authors could address this by considering a high-dimensional setup with $n / p \rightarrow c$ and adjusting the assumptions accordingly.

**Main Review:**

I found the paper to be well-written and easy to follow. However, I found the results to be non-surprising if not straightforward interpretations of well-known properties of ridge regression, and as such I believe that this contribution is not sufficiently significant for publication. I will detail my comments below.

On proposition 1 and 2. Propositions 1 and 2 are immediate from straightforward properties of the least-squares estimator: proposition 1 by noting that convex functions are lower-bounded by all their sub-tangents, and proposition 2 by the classical result that the specified quantities do not affect the loss up to scaling (as proved by the authors).

On the form of the assumptions and statement of theorem 1. The statement of the assumptions and the theorem could be improved by stating theorem 1 as a quantitative bound over the quantities defined in the assumptions, as the current “asymptotic” setup is not necessary to obtain the result, and is somewhat unnecessarily confusing due to the fact that no particular sequence of problems is described.

On the “asymptotic” setup. The current setup considered by the authors (with the requirement that $n \rightarrow \infty$ for $p$ fixed) is of limited interest for the problem at hand. Indeed, in such a context, the importance of regularization is much diminished (as there is no bias-variance trade-off), and it is not clear to me that this corresponds to typical settings of interest for hyper-parameter selection. Instead, it seems that the high-dimensional asymptotic $n \rightarrow \infty$, $n / p \rightarrow c$ is more appropriate for this problem, and given the existence of consistency result for the LOOCV estimator [1], it is not unreasonable to expect that similar results for quasi-convexity exist.

On the title of the paper. I would suggest the authors amend their title to de-emphasize the optimization aspect and highlight their contribution to characterizing the shape of the LOOCV loss for ridge regression, as there are no difficulty optimization difficulties in the problems highlighted by the authors. Indeed, in models like the ridge or LASSO which contain a scalar tuning parameter (or perhaps a couple of tuning parameters), grid search is a perfectly appropriate strategy for minimization and used in practice. Additionally, I note that for many such models, obtaining the solution path is not necessarily more expensive than obtaining a single solution (e.g. see [2], section 2.5), and hence grid search does not necessarily entail a heavy computational burden.

[1] "Error bounds in estimating out-of-sample prediction error using leave-one-out cross-validation in high dimensions". Rad, Zhou, Maleki. AISTATS 2020

[2] "Regularization Paths for Generalized Linear Models via Coordinate Descent". Friedman, Hastie, Tibshirani. J. Stat. Soft. 2010

Edit after author response

I would like to thank the authors for their response addressing my questions in detail. Unfortunately, I overall still feel that this contribution is somewhat too minor for this venue. To clarify some of the points I made in the review:
- on proposition 1 / 2: as the authors correctly point out, I was sloppy in my description of the invariance. What I meant is that all of the described quantities can be accounted for through straightforward change of variables in the presentation of the loss (scaling, rotation of $\beta$, and global scaling of $\beta$ which corresponds to a scaling of $\lambda$). I do believe that these are well-known properties of the OLS / ridge estimator.
- on the asymptotics: although I agree with the authors that parametric asymptotics may be useful in giving indicative behavior for finite sample problems, I am not currently convinced that their exists a finite sample regime where i) regularization is necessary and ii) parametric asymptotics are a good description of the problem. See e.g. empirical and theoretical analysis presented in [1]. Similarly, believe that high-dimensional asymptotics would provide a much more faithful representation of the behavior in finite sample regimes of interest.

[1]: A modern maximum-likelihood theory for high-dimensional logistic regression, Sur and Candes. PNAS 2019


**Time Spent Reviewing:**

5

---

> ### Author Response · Authors · 2021-08-10
> **Reviewer response**
>
> We thank the reviewer for their thoughtful review and detailed comments. We hope that we have addressed them thoroughly in our general comment to all reviewers as well as in detailed responses below. We respectfully ask the reviewer to reconsider their score in light of our responses.
>
> **“Propositions 1 and 2 are immediate from straightforward properties of the least-squares estimator.”** We completely agree that Propositions 1 and 2 are straightforward to prove and are not necessarily surprising results; we include them since they are important for the development of later ideas in the paper. The reviewer states that Proposition 2 is especially trivial because it follows from “the classical result that the specified quantities do not affect the loss up to scaling.” We disagree on this point. Scaling Y does affect the LOOCV loss up to a scaling. Changing V does not affect the LOOCV loss at all. Scaling the singular values actually changes the LOOCV loss pointwise, but in a way that does not change the quasiconvexity of L. We are not aware of Proposition 2 being a classical result; in fact, we are not aware of any reference discussing the quasiconvexity of the LOOCV loss. But if the reviewer has a reference for this result, we would be happy to include it in a revision.
>
> **“On the form of the assumptions and statement of theorem 1.”** The sequence of problems that Theorem 1 is concerned with is any sequence of problems meeting Assumptions 1--4. We agree this could have been more clear and will clarify this in the statement of the theorem. The reason we use an asymptotic setup here is because it is not clear to us how to analytically work through certain aspects of the proof otherwise. For example, it is not clear how to control the behavior of $\lambda_Q$ in Lemma 3 in any kind of meaningful finite-sample manner. As another example, we feel the same is true of $\lambda_Q’$ in in Lemma 4. But we agree that interpretable finite-sample results would be excellent to have. Can the reviewer clarify why they feel the asymptotic setup is not necessary to obtain our results?
>
> **“On the “asymptotic” setup. The current setup considered by the authors (with the requirement that  $n \to \infty$ for p fixed) is of limited interest for the problem at hand.”** We agree that when D=fixed and N has been taken to infinity, there is no need for regularization. Indeed, this is why we are sure to run experiments with N only moderately larger than D (lines 244-245). But that doesn’t mean it is useless to study the asymptotic regime of N tending towards infinity with D fixed. Otherwise one might similarly argue the following: because there is no uncertainty in statistics when N is infinite and D=fixed, the fixed-dimensional central limit theorem has no value. But of course the value is that we hope the asymptotic conclusions of the CLT hold for modest N; and empirically they do, so the CLT sees wide use. Similarly, we might hope that our asymptotic conclusions hold for modest N, and empirically, in Section 6, we show they do.
> The reviewer also suggests that we consider the case of $D / N \to c < \infty$. We think this is an interesting direction and believe that our major results should in fact go through in this case if our assumptions still hold and $c < 1$. However, we are uncertain as to whether our assumptions are reasonable in this setting (especially Assumption 3). We will be sure to investigate this empirically and revise our text if we find our assumptions are correct. We thank the reviewer for the suggestion.
>
> **“in models like the ridge or LASSO which contain a scalar tuning parameter (or perhaps a couple of tuning parameters), grid search is a perfectly appropriate strategy for minimization and used in practice”** We did not mean to imply that grid search is advantaged or disadvantaged as a method for optimizing the LOOCV loss when compared to gradient descent. Neither grid search nor gradient descent solves the problem of global optimization in the presence of local optima, even in one dimension. In particular, grid search can verify a local optimum by finding a region of decrease and then increase. But if local optima might exist, performing grid search up to some lambda_max is not guaranteed to find a global optimum; what if the global optimum lies beyond lambda_max? Additionally, even if lambda_max upper bounds the global optimum, a particularly sharp global optimum necessitates a very fine grid. So it does seem that there are serious difficulties for grid search (as well as for gradient descent) implied by our results.

---

### Official Review · Reviewer_zWsE · 2021-07-17

**Rating:** 6
**Confidence:** 4

**Summary:**

### Update

I have read the author response; I'd like to thank the authors for the detailed comments and clarifications.

- I am convinced from the authors' response that all of the theoretical issues I raised in my review can be fixed/clarified. Indeed, the response above does exactly this and need only be integrated into the main paper.

- I like the proposed experiments illustrating that GD on the LOO-CV loss can get stuck in local minima. This will be a nice demonstration of why the theoretical results matter in practice --- especially if the resulting test accuracies are quite different between the local min and the global solution.

In light of the response, I have decided to increase my score to a 6.

==========

The authors conduct a theoretical and empirical investigation of the _quasi-convexity_ of the leave-one-out cross validation (CV) metric for $\ell_2$-regularized least-squares regression (aka ridge-regression).
The main theoretical results are: i) counter-examples for general (quasi-)convexity of the CV metric, ii) an enumeration of dataset properties which do not effect quasi-convexity (scaling of the singular values of $X$, scaling of the targets $Y$, and the right singular vectors), and iii) sufficient conditions for quasi-convexity to hold.
The empirical component of the submission validates the sufficient conditions for quasi-convexity --- thus confirming the theoretical developments --- and examines sensitivity of the main result to their violation.
Interestingly, these experiments show that quasi-convexity is highly sensitive to the singular-value spectrum, but relatively robust to irregularity in the residuals and left singular vectors.

**Ethical Concerns:**

None.

**Limitations And Societal Impact:**

Yes, this is a theoretical paper with limited negative societal impacts.

**Main Review:**

# Update

The authors provided detailed clarifications on the theoretical issues that I raised in my review. In addition, they have promised to provide additional experiments illustrating the practical consequences of quasi-convexity of the LOO-CV loss. As as result, I have increased my score to a 6.

# Detailed Comments

This submission develops an interesting perspective on model selection for $\ell_2$ least-squares by CV.
The main motivation is direct optimization of the CV loss, either by grid-search or by gradient method.
Such optimization comes with optimality guarantees if the CV loss is quasi-convex, leading to the main effort of the paper.
I think this is a great problem that moves past ERM towards more direct optimization of the risk.

I believe the theoretical work is sound, but not groundbreaking.
Some areas of the proofs are unclear and missing details/steps which would both aid the reader and make verifier the results much easier.
See "Theory" below for further discussion of these issues.
The submission does not develop any tools/proof techniques that I think are immediately useful elsewhere, since most of the analysis proceeds by comparisons with quadratic functions.
Furthermore, use of such proof techniques means the result is probably hard to extend beyond least-squares regression, where the gradient of the CV loss will deviate more significantly from a quadratic.

The experiments are do a good job of confirming the theoretical developments.
In particular, it really does seem that regularity of the singular value spectrum is critical for quasi-convexity.
The major downside of the experiments is that they are limited to sensitivity analysis;
no progress is made towards using quasi-convexity for model selection.
In other words, the consequences of the theoretical results are not explored at all.

Overall, I feel this submission is borderline.
It makes a nice, but small contribution to our understanding of cross validation that I feel is unlikely to effect current practice.
I am willing to increase my score if the authors can clarify some of their proofs and provide (or just detail) some interesting experiments showing how we could use their results in practice.
Some potential areas to explore could be:

1. Practical ways to ensure the singular value spectrum is near-uniform.
2. A demonstration of gradient-descent on the CV loss for a quasi-convex instance. How does the computational cost compare to grid-search?

## Writing

The writing in the main text is polished.
The appendix has many small typographic mistakes --- see "Minor Comments" below.

## Theory

### Proposition 1

The proposition is correct, but I can't follow the proof as given.
How is the inequality $\mathcal{L}(\lambda) \geq \lambda \delta$ established?
The first-order definition of convexity gives
$$
\begin{aligned}
\mathcal{L}(\lambda) &\geq \mathcal{L}(\lambda^*) + \mathcal{L}'(\lambda^*)(\lambda - \lambda*)\\\\
&\geq \mathcal{L}(\lambda^*) + \delta(\lambda^* + 1 - \lambda*)\\\\
&= \mathcal{L}(\lambda^*) + \delta\\\\
&= \delta.
\end{aligned}
$$
which is not stated inequality.
Perhaps the intention was to define $\lambda = \lambda^* + \lambda'$, which then gives,
$$
\mathcal{L}(\lambda) \geq \mathcal{L}(\lambda^*) + \delta \lambda' \geq \delta \lambda'.
$$
This diverges as $\lambda' \rightarrow \infty$ (and thus $\lambda \rightarrow \infty$), as desired.

### Remark 1

Why is Condition 1 necessary here? It is not used in the proof of Prop. 2, which seems sufficient to make Remark 1 WLOG.

### Proposition 3

Please add a line to the proposition statement clarifying that it is proved when $\hat \theta$ is the solution the unregularized problem ($\lambda = 0$)?
The claimed equality $\hat E^\top U \hat \theta$ is easily verified for $\lambda = 0$, but does not hold for $\lambda > 0$ even if $S = I$.
Also, I assume that it is in fact $\hat E$ which is intended here, rather than $E$, as written in Line 536.
Similar clarification would be appropriate in Line 566.

### Lemma 2

The inequality in the first display of the proof is should be $\leq$ instead of $\geq$.
That, or negative signs are missing on both signs of the inequality.

### Equations 13 and 19:

These equations are difficult to check because all of the calculations have been omitted.
I suggest you include some additional details in the derivation (both for reviewing and for futer readers).

### Lemma 3

**Line 582**: How did you conclude that $\xi_{\cdot, 3} < 0$?
Clearly the first term is negative, but the second term has sign which depends on $\hat \epsilon_n \langle u_n, \hat \theta_n$.
Moreover, as $N \rightarrow \infty$ it is not clear which of the two terms decays faster.

**Display after Line 582**: this only includes the first term of $\xi_{\cdot, 3}$.
How was
$$
\sum_n \left(\frac{1}{(1 - \|u_n\|^2)^3} - 1\right) \left((1 - \|u_n\|^2) \hat \epsilon_n \langle u_n, \hat \theta \rangle \right)
$$
controlled?
Proposition 3 shows the sum $\sum_n |(1 - \|u_n\|^2) \hat \epsilon_n \langle u_n, \hat \theta \rangle|$ is $o(1)$.
I suppose that Assumption 3 implies $\left(\frac{1}{(1 - \|u_n\|^2)^3} - 1\right) \rightarrow 1$ as $N \rightarrow \infty$; upper-bounding this by $\left(\frac{1}{(1 - \|u_{max}\|^2)^3} - 1\right)$ and pulling it out of the sum should allow you to use Prop. 2 to obtain that this term is $o(1)$ as well.
Is this the intended argument?
Please either include a similar argument in the manuscript or provide an explanation for your reasoning.


### Lemma 4

**Line 603**: How did you conclude that $\lambda_Q' = \lambda_Q + O(1)$?
I see that
$$
\lambda_Q' = \lambda_Q + \frac{b + \xi_{\cdot, 2} - [\xi_{\cdot, 2}^2 - 4 \xi_{\cdot, 1} \xi_{\cdot, 3}]^{1/2} + [b^2 + 4 \xi_{\cdot, 1} c]^{1/2}}{2 \xi_{\cdot, 1}},
$$
where $c \geq - \xi_{\cdot, 3}$ for sufficiently large $N$.
Thus, $[b^2 + 4 \xi_{\cdot, 1} c]^{1/2} - [\xi_{\cdot, 2}^2 - 4 \xi_{\cdot, 1} \xi_{\cdot, 3}]^{1/2} \geq 0$ for large $N$ and
the goal becomes showing $[b + \xi_{\cdot, 2}] / 2 \xi_{\cdot, 1} \in O(1)$?

I guess an alternative is to show,
$$
\frac{- [\xi_{\cdot, 2}^2 - 4 \xi_{\cdot, 1} \xi_{\cdot, 3}]^{1/2} + [b^2 + 4 \xi_{\cdot, 1} c]^{1/2}}{2 \xi_{\cdot, 1}} \in O(1).
$$
It seems that $\xi_{\cdot, 3} \in o(1)$ by Assumptions 1, 3, in which case the difference is $O(1)$;
did you establish that $\xi_{\cdot, 1} \in O(1)$ at some point?

Regardless of the above, please provide more details for this step of the proof.

## Experiments

### Figure 3

The top-left hand plot is great confirmation of the theoretical developments. Very nice!

How did you check that the datasets were quasi-convex?
This could be accomplished by evaluating the CV loss on a fine grid and checking for convexity of the sub-level sets, but this is somewhat unattractive, since there may always be false negatives.
I am curious if the authors have a better procedure.

## Minor Comments

- Figure 1: the labels and other figure elements are too small to read.

- Remark 1: $Y$ is a vector on the $N - 1$ dimensional unit sphere since $Y$ is $N$ dimensional.

- Figure 2: "How does quasiconvexity depend on Y and U , for data with N = 3 and D = 2?" --- rhetorical questions are bad style.
    Try something like "Dependence of quasi-convexity on Y and U for data with N = 3 and D = 2".

- Line 124: I would emphasize that this is the "truncated" or "compact" SVD.
    I was a bit confused later, since $\|u_n\| = 1$ for the "full" SVD and Assumption 3 is then absurd.

- Line 137: $U^\top 1 = 0$ is only a sufficient condition for $X$ to have mean-zero columns.
    If $X$ is not full rank, then some singular values may be zero and $U^\top 1 \neq 0$ can hold.
    However, it is a necessary condition _if_ you further assume $X$ is full rank.

- Line 140: what do you mean by "normalized" with respect to the singular values?
    The diagonal values of $X^\top X$ are exactly $N$ because of the unit-variance assumption.
    Since the singular values of $X$ are the square-roots of the eigenvalues of $X^\top X$, it holds that $\sigma_0(X)^2 \leq N^2$ (consider a simple bound based on the trace) and $\sigma_N(X)^2 \geq 1$.
    Moreover, I don't believe you can conclude anything stronger on the distribution of the spectrum, except that $\sum_{i} \sigma_i(X)^2 \leq N^2$.
    I would not usually call this normalized, which I think of as between $0$ and $1$.

- Line 170: There has been no assumption that $X$ is full rank until now (see my earlier comment).
    In fact, is the only time "full-rank" is used in the manuscript as far as I can tell.

- Line 185: It is not true that $\|E\|^2 / N \in O(1)$ almost surely.
    That is, for every finite $N$, the event $\mathcal{E} = \|E\|^2 \geq N \log(N)$ has non-zero probability.
    To see this, consider that the $e_i$ are independent, so a lower-bound on the probability of $\mathcal{E}$ is $\prod_i^N P(e_i^2 \geq \log(N))$, which is clearly non-zero for every finite $N$.
    Thus, it is only correct to say that $\|E\|^2 / N \in O(1)$ with _high probability_.

- Theorem 1: it is an abuse of notation to write $S \in \Delta$, since $S$ is a matrix.
    I suppose the intent is $\text{Diag}(S) \in \Delta$.
    Note that this notation is used frequently throughout.

- Figure 3: the labels and other figure elements are somewhat small and low-resolution.

- Eq. (5) (Appendix): Why does the intermediate expression have a $c^2$ multiple? The $c^2$ terms from the numerator and denominator cancel. Maybe this is an oversight?

- Eq. (6) (Appendix): This expression is missing an inverse sign.

- Line 495 (Appendix): again, $Y$ is vector on the $N - 1$ dimensional unit sphere, not $D$.

- Line 608 (Appendix): "We first give state a theorem..." -> "We first state a theorem..."

- Line 612 (Appendix): The definition of $s_D$ should come before or in the statement of Theorem 2.


**Time Spent Reviewing:**

8

---

> ### Author Response · Authors · 2021-08-10
> **Reviewer response**
>
> We are very grateful to the reviewer for their especially detailed comments and attentive reading of our proofs and generally for looking so carefully through our paper. We address general points in a comment to all reviewers and more specific points below; we leave the proof responses until the end to collect them in one place.
>
> **To check the singular value spectrum is near uniform.** We consider three ways (A, B, C below) to interpret this comment for completeness.
>
> (A) Can we modify the design matrix? The design matrix we are given presumably corresponds to a practically meaningful regression, so we assume we cannot modify the design matrix without fundamentally changing the problem itself.
>
> (B) How should we computationally access the spectrum of X? There are many ways to do this, and we will include a discussion of these options in a revision:
> (i) When N and D are small (as in our examples), you can directly compute the SVD to verify the spectrum is near uniform.
> (ii) When D is small (and for any N), you can form X^(T)X and estimate its largest and smallest eigenvalue (and thereby bound the non-uniformity of the spectrum) efficiently via Rayleigh quotient iteration (see Trefthen and Bau 1997).
> (iii) If N>>D, you may wish to avoid forming the Gram matrix. In this case, you can use randomized sketching to get a $(1\pm \epsilon)$ approximation to the singular values in O(min(N*D*log(D)+D^(3),nnz(X)+D^4)) time [Woodruff 2014].
> (iv) If N~D are both large, you may wish to use spectral density estimation [Lin et al. 2016], which gives an estimate for the density of a matrix’s eigenvalues, and has been shown to scale to large problems [Ghorbani et al. 2019, Yao et al. 2020].
>
> (C) Given the spectrum of X, how can we know it is close enough to uniform for Theorem 1 to hold? Trying to answer this question is the purpose of our first experiment in the top-left of Figure 3. Here, we see that, for even modest $N$, as long as $\|S - 1\|_1 <= 10$, we can expect no non-quasiconvexity.
>
> **A demonstration of gradient-descent on the CV loss for a quasi-convex instance. How does the computational cost compare to grid-search?** We first want to clarify that our results are not necessarily related to a tradeoff between grid search and gradient descent. As we mention in our overview comment, both optimization algorithms can suffer in the presence of local optima, so we see our results as describing overall difficulties with optimizing the LOOCV loss, rather than difficulties for any one particular algorithm. For experiments comparing grid search and gradient descent for hyperparameter tuning, we point to Pedregosa 2016 (see the main text for a reference) for an example on $\ell_2$ regularized problems. Their finding is that gradient descent can outperform grid search in terms of computational complexity. Nonetheless, we will include experiments illustrating both gradient descent and grid search in the presence of quasiconvexity and non-quasiconvexity; e.g., in the latter case, we can set up these experiments to illuminate the pitfalls that both methods can face in the presence of local optima.
>
> **“The submission does not develop any tools/proof techniques that I think are immediately useful elsewhere, since most of the analysis proceeds by comparisons with quadratic functions”** We agree with this comment to an extent. However, we will point out that this is a fairly common occurrence when analyzing a new sort of problem. As one example, most of the initial theory surrounding the Lasso [Knight and Fu, 2000, Wainwright 2009] was fully customized to linear regression with $\ell_1$ regularization. Not until later were the underlying proof techniques generalized to other generalized linear models [Negahban et al. 2009]. It is our hope that some of the key ideas in our proofs will carry over to other models -- say, via reduction to a quadratic function plus a correction term due to deviations from linear regression.
>
> **“The major downside of the experiments is that they are limited to sensitivity analysis; no progress is made towards using quasi-convexity for model selection. In other words, the consequences of the theoretical results are not explored at all.”**
> We see our principal goal as understanding the practicality of CV loss minimization procedures for minimizing out-of-sample loss -- and in particular understanding whether local optima in the CV loss may impede test loss minimization. We do not interpret these goals as a sensitivity analysis.
>
> We will be sure to make the implications of our work for model selection more explicit in a revision. As one example, consider Figure 1; in the Wine dataset, the two local optima are separated by orders of magnitude. Thus, either gradient descent or grid search (e.g. with a too-small grid) might return a model substantially different from the global optimum. On the other hand, if quasiconvexity holds, then essentially any optimizer -- including gradient descent and grid search -- can succeed in finding the global optimum. In this case, previous work (e.g. Pedregosa (2016) or the extensive applied literature using grid search) has shown that the LOOCV loss can be efficiently optimized. See also our discussion in the overview comment.
>
> **How did you check that the datasets were quasi-convex?** To check for quasiconvexity, we did in fact evaluate L on a dense grid, as the reviewer suggests. We then examined sign changes in the pairwise differences L(lambda + delta) - L(lambda). This is not entirely non-trivial to implement in a numerically stable way; we will be sure to include this information in an appendix. Also we will be sure to add a discussion of the pros and cons of this procedure; note that detecting non-quasiconvexity is conclusive in this procedure but it is possible to miss some forms of non-quasiconvexity (a point that relates to our general grid search discussion).
>
> **References:**
> K. Knight and W. Fu. Asymptotics for Lasso-type estimators. The Annals of Statistics. 2000.
>
> S. N. Negahban, P. Ravikumar, M. J. Wainwright, and B. Yu. A Unified Framework for High-Dimensional Analysis of M-Estimators with Decomposable Regularizers. Statistical Science. 2012.
>
> M. J. Wainwright. Sharp Thresholds for High-Dimensional and Noisy Sparsity Recovery Using $\ell_1$-Constrained Quadratic Programming (Lasso). IEEE Transactions on Information Theory. 2009.
>
> Ghorbani, Behrooz, Shankar Krishnan, and Ying Xiao. "An investigation into neural net optimization via hessian eigenvalue density." In International Conference on Machine Learning, pp. 2232-2241. PMLR, 2019.
>
> Lin, Lin, Yousef Saad, and Chao Yang. "Approximating spectral densities of large matrices." SIAM review 58, no. 1 (2016): 34-65.
>
> Trefethen, Lloyd N., and David Bau III. Numerical linear algebra. Vol. 50. Siam, 1997.
>
> Woodruff, David P. "Sketching as a Tool for Numerical Linear Algebra." Foundations and Trends® in Theoretical Computer Science 10, no. 1–2 (2014): 1-157.
>
> Yao, Zhewei, Amir Gholami, Kurt Keutzer, and Michael W. Mahoney. "Pyhessian: Neural networks through the lens of the hessian." In 2020 IEEE International Conference on Big Data (Big Data), pp. 581-590. IEEE, 2020.

---

> > ### Comment · Reviewer_zWsE · 2021-08-10
> > **Author Response**
> >
> > Thank you for the detail comments and clarifications in response to my review.
> >
> > **Theory**: I am convinced from the authors' response that all of the theoretical issues I raised in my review can be fixed/clarified. Indeed, the response above does exactly this and need only be integrated into the main paper.
> >
> > **Experiments**: I like the proposed experiments illustrating that GD on the LOO-CV loss can get stuck in local minima. This will be a nice demonstration of why the theoretical results matter in practice --- especially if the resulting test accuracies are quite different between the local min and the global solution.
> >
> > In light of the response, I have decided to increase my score to a 6.

---

> ### Author Response · Authors · 2021-08-10
> **Clarifications on proofs**
>
> We give clarifications on our proofs in response to the reviewer's comments here.
>
>
> **“Proposition 1… Perhaps the intention was to define… ”** Thank you for pointing this out. Yes, this is exactly what we meant to define. We will correct this in a revision.
>
> **Why is Condition 1 needed for Remark 1?** We need Condition 1 so that Y is a vector on the (N-2)-sphere. Proposition 2 only shows that Y can be taken to be a unit vector; it is the additional constraint from Condition 1 that Y be zero-mean that allows us to reduce its dimension further and take it to be on the unit (N-2)-sphere. This is why in later parts of the paper we write that Y is a vector on the unit (N-2)-sphere.
>
> **Comments on Proposition 3.** The reviewer is correct that a hat is missing from the E here. We will correct this. We do note that we only define $\hat\theta$ as the unregularized estimate ($\lambda = 0$) in Definition 2. We will recall this definition in the Appendix for easier reading.
>
> **On Lemma 2.** The missing negative signs are indeed a typo here; the two sides of this display, as well as the following one, should all have negative signs. Thank you for catching this!
>
> **Equations 13 and 19 lack derivations.** We omitted these derivations because they are somewhat tedious calculus/algebra. In hindsight, though, we agree these equations should have derivations; we will be sure to show the intermediate steps in a revision.
>
> **On the display right after line 582, where the second part of $\xi_{\cdot,3}$ is omitted.** Yes, this is exactly right! The intent was to have Lemma 2 bound both parts of this quantity and refer to Lemma 2 on line 583 to conclude that $\delta(0) = o(1)$. We will fix this in a revision.
>
> **On Lemma 4.** First, we can show that $\xi_{\cdot, 1} = \Theta(1)$ and is positive by Assumption 4. To see this:
> \begin{align*}
> 	\xi_{\cdot, 1} &= \| \hat\theta \|^2 - \sum_n \| u_n \|^2 ( \langle u_n , \hat\theta \rangle + \hat\varepsilon^2_n ) + o(1) \\\\
> &\geq  \| \hat\theta \|^2 - \sum_n \| u_n \|^2 ( \langle u_n , \hat\theta \rangle + 2 \hat\varepsilon^2_n ) + o(1) \\\\
> &= \Theta(1),
> \end{align*}
> Where the final equality holds by Assumption 4. With this, the proof follows more or less the reviewer suggests:
> $$
> 	\lambda_Q - \lambda_Q' = \frac{b_\cdot + \xi_{\cdot, 2} + [b_\cdot + 4\xi_{\cdot, 1} ((1-\|u_n\|^2) \xi_{\cdot, 1} - 3\xi_{\cdot, 3} ) ]^{1/2} - [ \xi_{\cdot, 2} - 4 \xi_{\cdot, 1} \xi_{\cdot, 3}]^{1/2}}{\xi_{\cdot, 1}}.
> $$
> As we know the denominator is positive and $\Theta(1)$, we just need to show the numerator is $\Theta(1)$. Using $b_\cdot \geq 0$ and Assumption 3, the numerator satisfies:
> $$
> 	\geq \xi_{\cdot, 2} + [4\xi_{\cdot, 1}^2 - 4\xi_{\cdot, 1}^2 O(N^{-p}) - 12 \xi_{\cdot, 3} \xi_{\cdot, 1} ]^{1/2}  - [\xi_{\cdot, 2}^2 - 4\xi_{\cdot, 1} \xi_{\cdot,3}]^{1/2}.
> $$
> Now, as $\xi_{\cdot, 1} = \Theta(1)$:
> $$
> 	\geq \xi_{\cdot, 2} + [\Theta(1) - 12\xi_{\cdot,3}\xi_{\cdot, 1}]^{1/2} - \xi_{\cdot, 2} - [ | 4 \xi_{\cdot,1} \xi_{\cdot,3} |]^{1/2}.
> $$
> Now, as $\xi_{\cdot, 3} = o(1) - \sum_n \| u_n \|^2 \hat\varepsilon_n^2$, either $\xi_{\cdot, 3} < 0$ or $\xi_{\cdot, 3}$ is positive and $o(1)$. So the numerator satisfies:
> $$
> 	\geq [\Theta(1) - 12\xi_{\cdot, 3} \xi_{\cdot, 1}]^{1/2} - - [ | 4 \xi_{\cdot,1} \xi_{\cdot,3} |]^{1/2},
> $$
> which is $\Theta(1)$ and positive. Thus $\lambda_Q' - \lambda_Q$ is $\Theta(1)$ and is positive. Thank you for pointing out that this portion was lacking in details; we will include these in a revision. We will additionally add a lemma collecting all of the relevant facts about the $\xi_{\cdot, i}$ and $a_\cdot, b_\cdot, c_\cdot$ to better organize our proofs.
>
> **What do we mean by “normalized” singular values (line 140).** We mean that the singular values have been normalized $s_d \to s_d / s_1$ so that they sit in $[0,1]$; this allows us to apply our results from Theorem 1. We will clarify this point, since, as the reviewer points out, we have not defined what we mean by “normalized” in this case. To clarify this normalization with respect to Condition 1: Condition 1 is standard data preprocessing that should always be done to $X$ and $Y$. However, Proposition 2 then shows that some *additional* pre-processing can be done to $X$ and $Y$ without affecting the quasiconvexity of $L$. We thus assume this additional pre-processing for the sake of studying $L$’s quasiconvexity.
>
> **The reviewer enumerates a number of additional minor issues.** We agree with nearly all of these (with the exception of $Y$ not being on the $N-2$-sphere, as discussed above). We thank the reviewer again for such a careful reading and will make these corrections / clarifications in a revision.

---

### Author Response · Authors · 2021-08-10
**Feedback to all reviewers and AC**


We thank the reviewers for their very detailed and careful reviews. Here we first briefly review our contributions and then respond to general reviewer feedback. We also reply to specific points in response to each reviewer comment.

In this paper, we show that the LOOCV loss even in $\ell_2$ regularized regression can have local optima in real problems, and that the existence of local optima is a complex function of the regression problem. We provide sufficient conditions for when local optima cannot occur and demonstrate the necessity of these conditions in experiments.

**Practical implications of our findings.** Previous work has shown that the global optimum of the CV loss matches the global optimum of the out-of-sample loss (possibly after simple corrections). We argue that it is important to understand if local optima exist in the CV loss since typical CV optimization algorithms might then not be guaranteed to find the CV global optimum and hence the out-of-sample global optimum. Clearly gradient descent can be fooled by local optima. But note that grid search can too. Grid search is necessarily over a (user-chosen) compact space and so might miss local optima outside of this space, and an insufficiently fine grid might miss steep “canyons” in the CV loss.

**Behavior of true out-of-sample loss.** Two reviewers point out an interesting potential missing link in our reasoning: namely, the discrepancy between a CV local optimum and the out-of-sample global optimum is less concerning if test loss is similar at all CV local optima. We will therefore include a figure similar to Figure 2 but showing the ratio of test losses between the global optimum of LOOCV and worst-case local optimum of LOOCV. We do suspect that the test loss can be substantially different at CV local optima; see for instance the orders-of-magnitude difference in values of lambda at local optima in Figure 1.

---

### Decision · Program_Chairs · 2021-09-27

**Decision:**

Accept (Poster)

**Comment:**

We thank the authors for the additional clarifications provided in the rebuttal. The reviewers were satisfied with the authors' response about theoretical concerns and the appreciated the additional experiments. All reviewers also agreed that the overall problem discussed in this work is of interest to the community and that the paper makes a novel contribution, albeit that its importance is narrow.